# Exploratory Retrieval-Augmented Planning For Continual Embodied Instruction Following

**Minjong Yoo[1], Jinwoo Jang[1], Wei-jin Park[2], Honguk Woo[1]***
[1]Department of Computer Science and Engineering, Sungkyunkwan University
[2]Acryl Inc.
mjyoo2@skku.edu, jinustar@skku.edu, jin@acryl.ai, hwoo@skku.edu

## Abstract

This study presents an Exploratory Retrieval-Augmented Planning (ExRAP) framework, designed to tackle continual instruction following tasks of embodied agents in dynamic, non-stationary environments. The framework enhances Large Language Models' (LLMs) embodied reasoning capabilities by efficiently exploring the physical environment and establishing the environmental context memory, thereby effectively grounding the task planning process in time-varying environment contexts. In ExRAP, given multiple continual instruction following tasks, each instruction is decomposed into queries on the environmental context memory and task executions conditioned on the query results. To efficiently handle these multiple tasks that are performed continuously and simultaneously, we implement an exploration-integrated task planning scheme by incorporating the information-based exploration into the LLM-based planning process. Combined with memory-augmented query evaluation, this integrated scheme not only allows for a better balance between the validity of the environmental context memory and the load of environment exploration, but also improves overall task performance. Furthermore, we devise a temporal consistency refinement scheme for query evaluation to address the inherent decay of knowledge in the memory. Through experiments with VirtualHome, ALFRED, and CARLA, our approach demonstrates robustness against a variety of embodied instruction following scenarios involving different instruction scales and types, and non-stationarity degrees, and it consistently outperforms other state-of-the-art LLM-based task planning approaches in terms of both goal success rate and execution efficiency.

## 1 Introduction

The application of Large Language Models (LLMs) in embodied AI is essential for harnessing common knowledge and immediately applying it to unseen tasks and domains without requiring additional training or data. Researchers are further enhancing task adaptation by integrating environmental information with the intrinsic common knowledge of LLMs [1, 2, 3, 4, 5, 6, 7]. This capability proves invaluable in fields such as home robotics and autonomous driving, where it enables embodied agents to learn across diverse instruction following tasks with minimal data requirements.

For embodied agents, these tasks are often not mere single, one-time instructions but are multiple and persistent, necessitating continuous access to environmental knowledge to reason and plan effectively for user needs. In such scenarios, the efficiency of repeatedly collecting environmental knowledge through interaction each time the agent plans can be suboptimal. Furthermore, there is a clear need to integrate and manage multiple user requirements effectively.

---

*Corresponding Author

38th Conference on Neural Information Processing Systems (NeurIPS 2024).

In this paper, we investigate *continual instruction following* for an embodied agent, where multiple tasks are contingent upon the real-time information of a continuously changing environment. This setup requires the agent to engage in ongoing exploration of the environment to adaptively respond to dynamic changes and fulfill the required tasks. To address the problem of continual instruction following, we present an exploratory retrieval-augmented planning (ExRAP) framework, designed to enhance LLMs' embodied reasoning capabilities by integrating environmental context memory.

In ExRAP, to improve effectiveness and efficiency in managing environment interaction and exploration loads for multiple embodied tasks, we employ an exploration-integrated task planning scheme, in which the information-based exploration is incorporated into the LLM-based planning process. This integrated planning scheme establishes a robust policy that balances the validity of the environmental context memory with the demands of environment interaction and exploration. We also devise a temporal consistency-based refinement scheme to ensure the robustness of memory-augmented query evaluation on environmental conditions. Through experiments with VirtualHome [8], ALFRED [9] and CARLA [10], we demonstrate that the ExRAP framework achieves competitive performance in both task success and efficiency compared to several state-of-the-art embodied planning methods, including ZSP [11], SayCan [1], ProgPrompt [3], and LLM-Planner [12].

Our contributions are summarized as: First, we propose a novel ExRAP framework, systematically combining LLMs' reasoning capabilities and environmental context memory into exploration-integrated task planning to tackle continuous instruction following tasks in non-stationary embodied environments. We also introduce two schemes tailored for exploration-integrated task planning in ExRAP, information-based exploration estimation and temporal consistency-based refinement on memory-augmented query evaluation. Finally, we demonstrate superior performance and robustness of ExRAP via intensive experiments with home robots and autonomous driving scenarios in VirtualHome, ALFRED, and CARLA.

## 2   Related work

**Embodied instruction following.** Embodied instruction following involves executing complex tasks based on an understanding of embodied knowledge. This aims to grasp various aspects of the physical environment including objects, their relations, and dynamics, and to plan appropriate sequences of actions or skills to complete the tasks specified by instructions successfully. In the area of task planning, there have been many works to combine LLMs' reasoning capabilities with environmental characteristics. Recent research explored the utilization of skills' affordances to compute their values [1, 2], implemented code-driven policies [3, 4], and generated reward functions [13, 14], while highlighting the use of LLMs' enhanced abilities in task planning. Moreover, LLM-driven environment modeling approaches, utilizing LLMs' common knowledge and reasoning about real-world objects, have been introduced [5, 6, 7]. These LLM-based approaches to task planning have been applied across a range of embodied instruction following tasks, facilitated by repeated interactions with environments, humans, or other agents [15, 16, 12, 17, 18, 19].

While these approaches underscore the versatility and depth of LLMs for task planning and embodied agent control, they often rely on a non-systematic integration of observations to the LLMs' reasoning process. Furthermore, they rarely consider continual instruction following scenarios, where an agent should handle a set of instructions continuously, adapting to real-time environmental conditions. In contrast, our work differentiates itself by incorporating agents' exploration capabilities, which are guided by information gains, into continual instruction following.

**Retrieval-augmented generation for LLM.** Research in retrieval-augmented generation (RAG) focused on efficiently executing tasks by sourcing and utilizing task-related information from databases. In particular, enhancing the performance of retrieval, which suggests relevant data when an LLM requires specific knowledge for the tasks, involves training retrievers [20, 21, 22, 23], fine-tuning the LLM to adapt the RAG process [24, 25], or exploiting the LLM itself for dynamic query reformation [26, 27]. In the area of embodied task planning, recent studies adopt the integration of RAG with task-specific demonstrations [12]. Our work also uses RAG for embodied task planning, but it uniquely emphasizes its dynamic aspects. For continual instruction following, we prioritize the relevance and significance of the agent's skills, not only to perform tasks but also to ensure continuous and efficient synchronization of its environmental memory with changes in the environment.

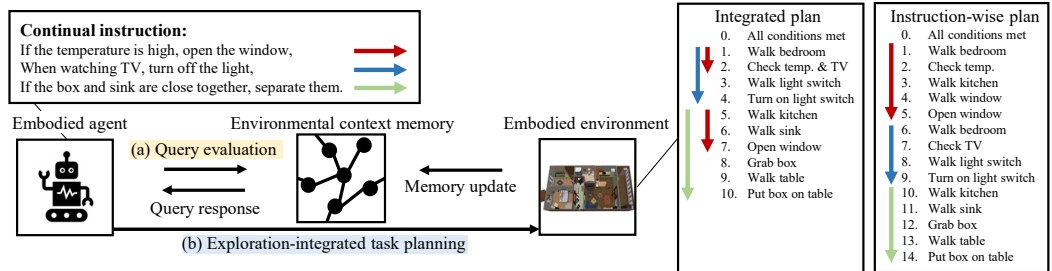

Figure 1: Concept of ExRAP. In the embodied environment, this framework manages continual instructions, a set of instructions for embodied instruction following tasks that are conducted continuously and simultaneously. At each step, it operates through (a) memory-augmented query evaluation, and (b) exploration-integrated task planning coupled with environmental context memory updates (as shown in the left side of the figure). By performing this integrated plan in conjunction with the memory, the ExRAP framework achieves more efficient task execution in response to the continual instructions, compared to the instruction-wise planning (as in the right side of the figure).

**Exploration in reinforcement learning (RL).** In the field of RL, exploration methods, designated to efficiently gather environmental information, have been a focus of research. Strategies were developed, that prioritize the exploration of new environmental information, by offering intrinsic rewards [28, 29] or through navigation schemes derived from offline demonstrations [30, 31, 32]. Particularly, in DREAM [33], an exploration policy is formulated to adapt to varying conditions by using data gathered during initial exploration episodes. This policy is optimized by maximizing the information gain to better adapt to changes. Our work adapts this mutual information-based exploration strategy with RAG for embodied instruction following. Our proposed framework is the first to implement exploration-integrated task planning with LLMs, enabling the efficient execution of continuously-performed instructions and facilitating the dynamic adaptation to changing environmental conditions.

## 3 Approach

### 3.1 Continual instruction following

We consider a set of instructions for embodied tasks that are continuously and simultaneously conducted based on specific environmental contexts. In Figure 1, the instructions entail conditional actions like "If the temperature is high, open the window" and "When watching TV, turn off the light." The agent continuously explores the environment to verify whether the conditions (e.g., temperature and TV status) are met. Upon confirmations, the agent executes the associated tasks within the environment. We refer to these scenarios, where *multiple* embodied tasks are conditioned on environmental contexts and conducted *continuously*, as continual instruction following. This concept aligns closely with continuous queries [34, 35] in database literature, which monitor updates of interest over time and return results when specific thresholds or conditions are met. This is in contrast to single in-situ instruction following, where each task is executed based on isolated, one-time directives.

For continual instruction following tasks with conditional instructions $\mathcal{I} = \{i_1, ..., i_M\}$, we consider a non-stationary embodied environment that changes over time. The conditions of continual instructions may or may not be satisfied over time, requiring continuous exploration in the environment. When certain conditions are met, the associated tasks should be performed promptly. For this continual instruction following tasks in the non-stationary environment, we evaluate agent performance in terms of task completion and efficiency. Our goal is to establish an embodied agent policy $\pi^*$ that maximizes the overall performance of continual instruction following tasks. Specifically, we formulate the reward as a combination of (i) the **task success rate SR** and (ii) the average **pending step PS**. SR is the rate of completed tasks whose conditions have been met, and PS is the average steps required to accomplish the task associated with instruction $i \in \mathcal{I}_C$ whenever the condition is met. For instructions $\mathcal{I}$ and timestep $t$, we then formulate the agent policy $\pi^*$ performing a skill upon observation $o_t$ as

$$\pi^* = \text{argmax}_\pi \left[ \sum_t \text{SR}(s_t, \pi(o_t, \mathcal{I})) + \mathbb{E}_{i \in \mathcal{I}_C}[-\text{PS}(\pi, i)] \right]. \tag{1}$$

Optimizing both SR and PS allows the agent not only to appropriately plan for multiple instruction following tasks but also to strategically integrate these instructions to improve overall efficiency. The resulting policy is able to minimize redundant skill executions by addressing multiple tasks in an integrated manner, taking into account the possible spatial and temporal overlap of the task requirements. For instance, as illustrated in Figure 1, an integrated plan achieves a pending step of 7, the average of required timesteps 7, 4 and 10 for the three instructions. This is significantly shorter than 9.2, the average of 5, 9, and 14 achieved by an instruction-wise plan.

## 3.2 Overall framework

To address the challenge of continual instruction following in a non-stationary embodied environment, we develop the ExRAP framework. It is designed to minimize the necessity for environmental interaction, by utilizing memory-augmented and exploration-integrated planning schemes while ensuring robust task performance.

In ExRAP, each conditional instruction $i \in \mathcal{I}$ is decomposed into two primary components: query $q$ and execution $e$. Queries function as conditions for task initiation and are evaluated against environmental information. Executions, on the other hand, involve physical interactive manipulations that are triggered based on the results of query evaluation. In a non-stationary environment, evaluating queries poses a unique challenge due to the need for the agent to continuously synchronize with constantly changing information. This synchronization often necessitates continual exploration, resulting in intensive interaction with the environment.

As described in Figure 1, ExRAP addresses this challenge through two components: (a) query evaluation using environmental context memory and (b) exploration-integrated task planning. In (a), the environmental context memory is established via a temporal embodied knowledge graph to effectively represent the dynamic environment. Augmented with this graph-based context memory, the LLM-based query evaluator responds to queries by checking if their conditions are met and provides confidence levels for these assessments. To address the information decay, which stems from synchronization uncertainty between the previously collected environmental context and the actual current state of the environment, we incorporate entropy-based temporal consistency refinement into the query evaluation process. In (b), ExRAP plans skills that are instrumental not only for achieving tasks from an *exploitation* perspective but also for boosting confidence in the query evaluations from an *exploration* perspective. To effectively plan skills that balance both perspectives, we integrate the exploitation value of skills, which is derived from the in-context learning ability of LLMs, with their exploration value, which is determined through information-based estimation.

## 3.3 Memory-augmented query evaluation with temporal consistency

We represent both the environmental context memory and the observations perceived by the agent using a temporal embodied knowledge graph (TEKG), where the memory is established through the accumulation of these observations. Queries, derived from given instructions, are evaluated against the context memory, with consideration for inherent information decay within the previously accumulated data. The query evaluation procedure is described on the upper side of Figure 2.

**TEKG and retriever.** The TEKG comprises a set of quadruples $\tau = (se, re, te, t)$ consisting of source entity $se$, relation $re$, target entity $te$, and timesteps $t$. We represent the environmental context memory at a specific timestep $t$ within the TEKG, defined as

$$G_t = \{\tau_1, \tau_2, \cdots, \tau_N\} \text{ where } \tau_i = (se_i, re_i, te_i, t_i), \ t_i \leq t. \tag{2}$$

To integrate the current observation $o_{t+1}$ into previously established up-to-date memory $G_t$, we employ an update function $\mu$ as follows:

$$G_{t+1} = \mu(G_t, o_{t+1}) = \{\tau \in G_t \,|\, c(\tau, \tau') = 0, \ \forall \tau' \in o_{t+1}\} \bigcup o_{t+1}. \tag{3}$$

Here, $c$ is a function that detects the semantic contradictions between quadruples, such as when $\tau$ and $\tau'$ indicate that a TV is both "off" and "on". It returns 1 if there is a contradiction, otherwise 0.

Constructed memory $G$ serves as a knowledge database for continual instruction following, enabling the retrieval of environmental information related to specific task directives such as instructions, queries, and executions. Specifically, for language-specified task directives $\mathcal{L} = \{l\}$, the retriever $\Phi_R$

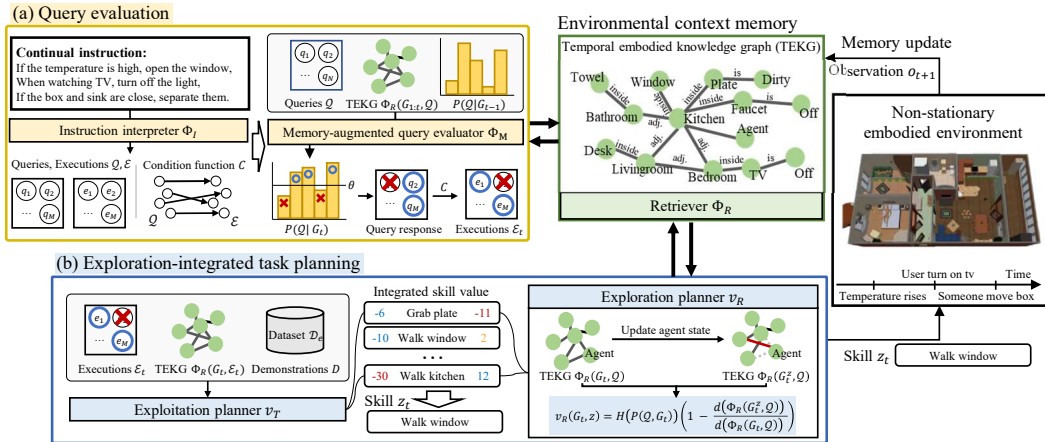

Figure 2: Overall procedures of ExRAP. (a) Query evaluation: The instruction interpreter $\Phi_I$ produces queries and executions, as well as a condition function from continual instructions. The memory-augmented query evaluator $\Phi_M$ then evaluates these queries probabilistically using the LLM with a retrieved TEKG from the environmental context memory. (b) Exploration-integrated task planning: The LLM-based exploitation planner $v_T$ estimates the value of skills based on their executions and relevant demonstrations in an in-context manner. Simultaneously, the exploration planner $v_R$ evaluates these skills using the subsequent TEKG through information-based value estimation. At each step, a skill is then selected based on the integrated skill value from the two estimations.

interacts with the memory $G$ and samples $k$ quadruples $\{\hat{\tau}_1, \cdots, \hat{\tau}_k\}$. The sampling is based on the multinomial softmax distribution, where the likelihood of retrieving a quadruple $\tau$ is determined by the highest sentence embedding similarity between $\tau$ and any $l$ in $\mathcal{L}$.

**Instruction interpreter.** The instruction interpreter $\Phi_I$ processes continual instructions $\mathcal{I} = \{i_1, ..., i_M\}$, translating them into queries $\mathcal{Q}$ and corresponding task executions $\mathcal{E}$:

$$\Phi_I(\mathcal{I}) = (\mathcal{Q} : (q_1, \ldots, q_M), \mathcal{E} : (e_1, \ldots, e_M), C) \text{ where } C(q_j) = e_j \text{ for } \forall j. \quad (4)$$

Here, $C$ is a conditional function that maps each query to its respective execution counterpart.

**Query evaluator.** The memory-augmented query evaluator $\Phi_M$ estimates the likelihood $P(q|G_t)$ of query $q \in \mathcal{Q}$ being satisfied, using the historical memory accumulated over time, denoted as $G_{1:t} = G_1 \cup ... \cup G_t$. Leveraging the memory-augmented LLM ($\Phi_{\text{LLM}}$), we develop the query evaluator $\Phi_M$ by incorporating the previous step's query evaluation $P(q|G_{t-1})$ and a prior of query evaluation $R(q|G_{t-1})$, which is defined in (7).

$$P(q|G_t) = \Phi_M(q, t, G_{1:t}, P(q|G_{t-1})) = \begin{cases} R(q|G_{t-1}) & \text{if } \hat{G}_{1:t} = \hat{G}_{1:t-1} \\ \Phi_{\text{LLM}}(q, t, \hat{G}_{1:t}, R(q|G_{t-1})) & \text{otherwise} \end{cases} \quad (5)$$

Here, $\hat{G}_{1:t} \sim \Phi_R(G_{1:t}, \{q\})$ is retrieved quadruples, and the prior $R(q|G_{t-1})$ is the retrospective query response evaluated at timestep $t$ using $G_{t-1}$.

Due to the inherent information decay in the memory over increasing timesteps, a decline in confidence should be considered for the likelihood estimation in (5). To address this, we incorporate an entropy-based **temporal consistency** as an intermediate step in query evaluation. Specifically, when using the memory from $G_{t-1}$, we posit that the entropy of the prior query response at timestep $t$ should be higher than at timestep $t-1$:

$$H(R(q|G_{t-1})) > H(P(q|G_{t-1})). \quad (6)$$

To enforce the temporal consistency, we compute the multiple query response priors using $\Phi_{\text{LLM}}$ and discard any responses that do not align with the consistency constraint. Specifically, if the entropy of each prior of query response is smaller than previous step $P(q|G_t)$, it is removed.

$$R(q|G_{t-1}) = \mathbb{E}_{\hat{G}_{1:t-1} \sim \Phi_R(G_{1:t-1}, \{q\})} \left[ \Phi_{\text{LLM}}\left(q, t, \hat{G}_{1:t-1}, P(q|G_{t-1})\right) \text{ with hold (6)} \right] \quad (7)$$

Then, we select a set of corresponding executions $\mathcal{E}_t$ that are likely to require manipulations in the environment, using a filtering threshold $\theta$.

$$\mathcal{E}_t = \{C(q)|q \in \mathcal{Q}, P(q|G_t) > \theta\} \tag{8}$$

### 3.4 Exploration-integrated task planning with information-based estimation

To facilitate integrated task planning for continual instructions, we devise exploitation and exploration planners. The former focuses on exploiting appropriate skills to complete tasks associated with given instructions using the LLM, and the latter focuses on exploring the environment to update the memory in a direction that maximizes information gain. We then integrate their plans to prioritize the next skills to be executed. The resulting planning directs to complete specific task executions $\mathcal{E}$, ensuring effective maintenance of the environmental context memory. This maintenance process involves synchronizing the memory with the current state of the environment. The exploration-integrated task planning procedure is described on the lower side of Figure 2.

**Exploitation planner.** Given the memory $G_t$ and a language description of a skill $z \in Z$, the exploitation planner $v_T$ is responsible for estimating the value of the skill with respect to its effectiveness in accomplishing the executions $\mathcal{E}_t$. To do so, we harness the retrieved memory-augmented LLMs along with their in-context learning capabilities. Specifically, under the assumption that we can access expert planning dataset $\mathcal{D}_e$, we retrieve demonstrations $D$ from $\mathcal{D}_e$ based on the graph similarity between the current observation $o_t$ and the observation within $\mathcal{D}_e$.

$$v_T(G_t, z) = \Phi_{\text{LLM}}(\mathcal{E}_t, \Phi_R(G_t, \mathcal{E}_t), D, z) \tag{9}$$

**Exploration planner.** In conjunction with the exploitation planner, the exploration planner $v_R$ is responsible for assessing the value of the skill with respect to its utility for reducing the response uncertainty of the query evaluator $\Phi_M$. This assessment is intended for efficient environmental exploration, thereby facilitating the swift and precise identification of query conditions and maintaining the memory up-to-date. Specifically, we define the exploration value of skill $z$ as the difference of mutual information for consecutive timesteps using the query evaluation result in (5):

$$v_R(G_t, z) = I(\mathcal{Q}; G_{t+1}) - I(\mathcal{Q}; G_t) = \sum_{q \in \mathcal{Q}} H(P(q|G_t)) - H(P(q|G_{t+1})) \tag{10}$$

where $I$ denotes mutual information and $G_{t+1}$ is the updated memory by execution of skill $z$.

Direct computation of $G_{t+1}$ is impractical without actual skill execution. Therefore, to evaluate the exploration value of skills before their execution, we focus on the entropy related only to the retrieved knowledge pertinent to the evaluated queries $\mathcal{Q}$. This approach is feasible under the mild assumption that the entropy of the query evaluator reaches zero (indicating no uncertainty), once the TEKG memory is fully synchronized with the environment. Thus, we approximate the exploration value as

$$v_R(G_t, z) = \sum_{q \in Q} H(P(q|G_t)) \left(1 - \frac{d(\Phi_R(G_t^z, \{q\}))}{d(\Phi_R(G_t, \{q\}))}\right) \tag{11}$$

where $d$ represents the average distance function between the retrieved quadruples and the current agent's entity in TEKG, and $G_t^z$ is the predicted partially updated knowledge graph after skill execution, where only quadruples containing the agent as an entity are altered. Note that the exploration value increases as the agent moves closer to the query-related environment parts on the graph through the skill execution. Consequently, the skill is selected by maximizing the integrated skill value, which is obtained by a weighted sum of exploitation and exploration values, as defined in (9) and (11) respectively:

$$z_t = \underset{z \in Z}{\text{argmax}}[w_T \cdot v_T(G_t, z) + w_R \cdot v_R(G_t, z)]. \tag{12}$$

## 4 Experiments

We evaluate ExRAP across various degrees of non-stationarity, scales of instructions, and instruction types. We also provide ablation studies and qualitative analysis. Further analysis is in Appendix D.

Table 1: Performance in VirtualHome, ALFRED, and CARLA w.r.t. non-stationarity. We use the 95% confidence interval, using 10 random seeds for VirtualHome and 5 random seeds for both ALFRED and CARLA.

| Model | Low non-stationarity | | Medium non-stationarity | | High non-stationarity | |
|---|---|---|---|---|---|---|
| | SR ($\uparrow$) | PS ($\downarrow$) | SR ($\uparrow$) | PS ($\downarrow$) | SR ($\uparrow$) | PS ($\downarrow$) |
| **Evaluation in VirtualHome** | | | | | | |
| ZSP | 20.59%±4.71% | 31.03±4.68 | 20.06%±1.93% | 32.06±4.66 | 17.28%±3.16% | 24.08±4.63 |
| SayCan | 35.12%±4.83% | 21.67±3.81 | 33.69%±5.36% | 21.81±4.14 | 27.33%±4.24% | 16.18±3.98 |
| ProgPrompt | 32.10%±4.41% | 18.84±4.08 | 30.51%±5.31% | 23.43±1.07 | 27.19%±2.99% | 18.60±4.22 |
| LLM-Planner | 40.97%±7.00% | 17.61±1.40 | 39.89%±4.52% | 15.93±2.13 | 34.60%±6.49% | 14.94±2.89 |
| ExRAP | **61.12%**±**7.03%** | **11.75**±**2.49** | **55.14%**±**6.59%** | **11.33**±**1.92** | **50.12%**±**5.70%** | **8.61**±**2.25** |
| **Evaluation in ALFRED** | | | | | | |
| ZSP | 18.22%±5.33% | 17.24±2.12 | 14.67%±6.18% | 20.83±3.63 | 9.56%±4.80% | 22.53±3.57 |
| SayCan | 45.67%±6.89% | 8.25±1.86 | 41.81%±7.64% | 8.39±3.55 | 35.79%±6.31% | 7.42±1.14 |
| ProgPrompt | 47.15%±1.17% | 9.81±2.14 | 35.62%±1.04% | 7.22±1.35 | 19.97%±0.80% | 7.52±2.45 |
| LLM-Planner | 58.44%±3.97% | 7.28±1.09 | 51.80%±3.79% | 7.28±1.05 | 35.76%±6.00% | 6.65±1.06 |
| ExRAP | **69.90%**±**1.47%** | **5.94**±**0.92** | **64.00%**±**5.07%** | **4.82**±**1.03** | **59.11%**±**2.48%** | **4.42**±**1.36** |
| **Evaluation in CALRA** | | | | | | |
| ZSP | 10.44%±1.03% | 29.35±7.21 | 6.89%±2.98% | 32.46±6.03 | 4.67%±1.17% | 33.00±1.40 |
| SayCan | 37.55%±4.74% | 20.73±5.36 | 35.11%±6.12% | 22.44±5.38 | 30.71%±5.28% | 21.71±2.01 |
| LLM-Planner | 50.83%±1.60% | 14.02±3.01 | 44.00%±0.70% | 14.39±1.94 | 41.58%±3.35% | 13.59±2.65 |
| ExRAP | **65.25%**±**7.47%** | **12.43**±**3.90** | **62.25%**±**6.72%** | **11.50**±**2.24** | **58.83%**±**10.08%** | **10.84**±**2.52** |

**Environments.** We evaluate ExRAP in the context of household planning and skill-based autonomous driving with VirtualHome [8], ALFRED [9], and CARLA [10], where we use 16 to 19 distinct instructions for continual instruction following tasks. Details are provided in Appendix A.

**Evaluation metric.** We employ two evaluation metrics for the objective specified in (1). The task success rate (**SR**) measures the proportion of completed tasks for continual instructions whose conditions are satisfied at each timestep. Given the continual instructions, the pending step (**PS**) represents the average number of timesteps required to complete the associated tasks from the moment the conditions of the instructions are actually satisfied in the environment. Note that the agent's detection time of such condition satisfaction may differ from its actual occurrence.

**Datasets.** We use 100 trajectories across 10 different environment settings in VirtualHome, and 50 trajectories in ALFRED and CARLA. These are used for in-context learning of the exploitation planner in ExRAP and the baselines. Note that we use different environment settings for evaluation.

**Baselines. ZSP** [11] is an LLM-based zero-shot task planner. Our experiments serve as a baseline to evaluate LLM-based task planning approaches. **SayCan** [1] is a state-of-the-art embodied agent framework, which integrates both language affordance scores derived from an LLM and embodied affordance scores learned through RL. In our experiments, we use the optimal affordance function for each environment. **ProgPrompt** [3] is a framework to enhance language models' capabilities in generating structured and logical outputs, by incorporating programming-like prompts. **LLM-Planner** [12] is a state-of-the-art embodied agent framework that utilizes LLMs' embodied knowledge to infer subtask sequences. It incorporates object detection information from the agent's interactions for enhanced task planning. To address continual instructions and adapt to non-stationary environments, we implement a variant of the LLM-Planner that infers skills in a step-wise manner.

## 4.1 Main results

**Non-stationarity.** Table 1 presents a performance comparison in terms of **SR** and **PS** in VirtualHome, ALFRED, and CARLA, respectively, under varying degrees of non-stationarity, where environment changes range from low to high. The higher degree of non-stationarity means the environment changes more rapidly, requiring the agent to focus more on environmental information to adapt effectively. ExRAP achieves superior performance across all degrees of non-stationary. Specifically, ExRAP demonstrates a performance gain in SR by 16.45% on average compared to the most competitive baseline, the LLM-Planner. Furthermore, ExRAP shows a reduction in PS by 3.40 on average compared to the LLM-Planner. Importantly, the advantage of ExRAP becomes more significant with

Table 2: Performance in VirtualHome w.r.t. instruction scale

| Model | Small continual inst. (=3) | | Medium continual inst. (=5) | | Large continual inst. (=7) | |
| | SR (↑) | PS (↓) | SR (↑) | PS (↓) | SR (↑) | PS (↓) |
| --- | --- | --- | --- | --- | --- | --- |
| ZSP | 33.11%±2.55% | 22.86±2.41 | 20.06%±1.93% | 32.06±4.66 | 7.43%±6.2% | 59.16±26.04 |
| SayCan | 40.58%±8.79% | 19.76±2.02 | 33.69%±5.36% | 21.81±4.14 | 23.04%±12.26% | 37.05±17.43 |
| ProgPrompt | 43.15%±3.22% | 19.99±1.59 | 30.51%±5.31% | 23.43±1.07 | 21.20%±7.45% | 29.00±1.27 |
| LLM-Planner | 49.28%±5.10% | 17.13±7.67 | 39.89%±4.52% | 15.93±2.13 | 31.82%±14.30% | 17.63±2.34 |
| ExRAP | **67.77%±4.56%** | **12.29±0.96** | **55.14%±6.59%** | **11.33±1.92** | **53.86%±8.59%** | **8.76±0.94** |

increasing levels of non-stationarity; the performance gap in SR between the LLM-Planner and ExRAP widens from 15.35% at low non-stationarity to 18.58% at high non-stationarity. Similarly, the gap in PS grows from an average of 2.96 to 3.73. This increase in performance can be attributed to ExRAP's strong ability to promptly discern newly satisfied conditions through exploration-integrated task planning, coupled with accurate memory-augmented query evaluation. This capability enables ExRAP to respond to rapid environmental changes effectively.

**Instruction scale.** Table 2 shows a performance comparison under the medium non-stationarity, as the scale of continual instructions increases. As the number of instructions grows, the agent needs to collect more knowledge and perform more tasks. ExRAP achieves an average gain in SR of 18.78% compared to the most competitive baseline, the LLM-Planner. Additionally, ExRAP reduces PS by an average of 6.31. ExRAP exhibits widening performance gaps as the complexity of tasks increases; the SR gap grows from 18.49% compared to the LLM-Planner with a small continual instruction scale, to 22.04% with a large continual instruction scale. Similarly, the PS gap expands from an average of 4.84 to 8.87. This performance difference highlights ExRAP's effectiveness in addressing multiple continual instructions simultaneously, demonstrating its robust integrated task planning capabilities. In ExRAP, the skill selection is guided by an integrated value derived from queries and executions, enabling the efficient handling of multiple instructions and concurrent memory updates.

**Instruction type.** We also test the applicability of ExRAP for three types of continual instructions in VirtualHome with medium non-stationarity. **Sentence-wise** type organizes each instruction into individual sentences. This is the default setting in our experiments. **Summarized** type condenses multiple instructions into a fewer number of sentences. **Object Ambiguation** type contains abstract forms of target objects such as "something to read.". Table 3 demonstrates superior performance of ExRAP for those types. ExRAP adapts RAG with the environmental context memory to interpret instructions and decompose them into queries and executions. This memory-augmented approach, retrieving and utilizing the information relevant to given instructions, enables effective grounding of continual instructions in different types within the environment.

Table 3: Performance w.r.t. instruction types

| Model | Sentence-wise | | Summarized | | Object Ambiguation | |
| | SR (↑) | PS (↓) | SR (↑) | PS (↓) | SR (↑) | PS (↓) |
| --- | --- | --- | --- | --- | --- | --- |
| SayCan | 33.69%±5.36% | 21.81±4.14 | 32.66%±4.29% | 22.13±5.67 | 23.89%±10.97% | 28.03±5.54 |
| LLM-Planner | 39.89%±4.52% | 15.93±2.13 | 37.19%±3.48% | 16.45±3.87 | 30.88%±5.20% | 19.09±4.92 |
| ExRAP | **55.14%±6.59%** | **11.33±1.92** | **53.11%±15.10%** | **10.42±4.28** | **50.26%±10.91%** | **13.51±6.07** |

**Qualitative analysis.** Figure 3 (a) and (b) compare the exploration strategies of ExRAP and LLM-Planner. Given multiple instructions, ExRAP demonstrates broader exploration. This reflects the exploration-integrated task planning in ExRAP, which rather focuses on the overall gain achieved from each exploration and skill execution.

## 4.2 Ablation study

We conduct several ablation studies for ExRAP in VirtualHome with medium non-stationarity.

**Temporal consistency.** We compare the performance of our ExRAP and its variant ExRAP-TC that performs query evaluation without temporal consistency-based refinement. As in Table 4, ExRAP outperforms ExRAP-TC by 15.56% on average. As observed in Figure 3 (a) and (c), with temporal consistency in query responses, information decay is effectively managed, leading to broader

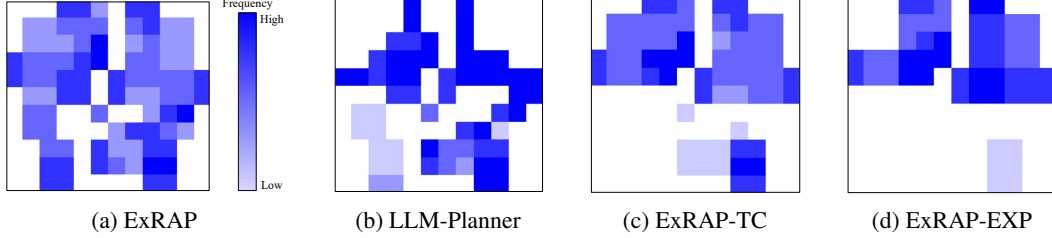

|  (a) ExRAP | (b) LLM-Planner | (c) ExRAP-TC | (d) ExRAP-EXP |

Figure 3: Knowledge exploration heatmap. Darker color represents high frequency in exploration.

exploration areas that accommodate different instructions simultaneously (i.e., in (a)). Otherwise, without temporal consistency, exploration tends to concentrate exclusively on specific knowledge where multiple queries overlap. This renders largely neglected areas experiencing significant decay, often leading to invalidated query evaluation (i.e., in (c)).

Table 4: Ablation for query evaluation with temporal consistency

| Model | Low non-stationarity | | Medium non-stationarity | | High non-stationarity | |
|---|---|---|---|---|---|---|
| | SR (↑) | PS (↓) | SR (↑) | PS (↓) | SR (↑) | PS (↓) |
| ExRAP-TC | $47.25\%_{\pm18.00\%}$ | $15.30_{\pm6.97}$ | $43.92\%_{\pm7.97\%}$ | $15.47_{\pm2.17}$ | $27.91\%_{\pm14.64\%}$ | $9.67_{\pm5.32}$ |
| ExRAP | $\mathbf{61.13\%}_{\pm\mathbf{13.76\%}}$ | $\mathbf{11.66}_{\pm\mathbf{3.93}}$ | $\mathbf{55.14\%}_{\pm\mathbf{6.59\%}}$ | $\mathbf{11.33}_{\pm\mathbf{1.92}}$ | $\mathbf{49.73\%}_{\pm\mathbf{8.88\%}}$ | $\mathbf{8.74}_{\pm\mathbf{2.74}}$ |

**Exploration strategy.** We compare the performance of our ExRAP and two variants ExRAP-LLM and ExRAP-EXP using different planning strategies. ExRAP-LLM directly employs the LLM as the exploration planner, which fully relies on the LLM's capability without information-based exploration, by prompting the LLM with specific exploration commands, e.g., "explore the home." On the other hand, ExRAP-EXP employs only the exploitation planner with specific exploration commands, used only when there are no executions. As shown in Table 5, ExRAP demonstrates higher performance of 26.76 and 17.46% on average than the two variants, respectively. As in Figure 3 (a) and (d), ExRAP-EXP exhibits reduced exploration capabilities, resulting in a narrower exploration area.

Table 5: Ablation for exploration-integrated task planning

| Model | Low non-stationarity | | Medium non-stationarity | | High non-stationarity | |
|---|---|---|---|---|---|---|
| | SR (↑) | PS (↓) | SR (↑) | PS (↓) | SR (↑) | PS (↓) |
| ExRAP-LLM | $34.23\%_{\pm24.57\%}$ | $15.17_{\pm5.11}$ | $29.42\%_{\pm25.63\%}$ | $10.78_{\pm3.84}$ | $21.81\%_{\pm14.06\%}$ | $14.65_{\pm2.40}$ |
| ExRAP-EXP | $43.33\%_{\pm11.28\%}$ | $13.07_{\pm4.75}$ | $40.68\%_{\pm9.77\%}$ | $13.03_{\pm3.01}$ | $29.36\%_{\pm14.35\%}$ | $12.16_{\pm3.86}$ |
| ExRAP | $\mathbf{61.13\%}_{\pm\mathbf{13.76\%}}$ | $\mathbf{11.66}_{\pm\mathbf{3.93}}$ | $\mathbf{55.14\%}_{\pm\mathbf{6.559\%}}$ | $\mathbf{11.33}_{\pm\mathbf{1.92}}$ | $\mathbf{49.73\%}_{\pm\mathbf{8.88\%}}$ | $\mathbf{8.74}_{\pm\mathbf{2.74}}$ |

**LLMs for planning.** While we tested several LLMs, ranging from relatively smaller models such as Gemma-2B [36] to larger ones such as Llama-3-70B [37], our experiments thus far have utilized LLaMA-3-8B for ExRAP and the baselines. In Table 6, we evaluate ExRAP and two baselines using different LLMs. Both LLM-Planner and SayCan exhibit improved performance with the larger Llama-3-70B, but experience a significant drop in performance with the smaller Gemma-2B model. Unlike those, ExRAP maintains robust performance even with the smaller model, highlighting the benefits of its memory-augmented, integrated planning approach.

# 5   Conclusion

We introduced the ExRAP framework to facilitate efficient integrated planning for multiple instruction following tasks, which are conducted continuously and simultaneously in the embodied environment. With the extended RAG architecture, the framework incorporates memory-augmented query evaluation and exploration-integrated task planning schemes, thereby achieving both efficient environment exploration and robust task completion. Via experiments conducted in VirtualHome, ALFRED, and CARLA, we demonstrated the robustness of ExRAP across various continual instruction following scenarios, specifying its advantages over other LLM-driven task planning approaches.

Table 6: Impact of different LLMs

| Model | Gemma-2B | | Llama-3-8B | | Llama-3-70B | |
|---|---|---|---|---|---|---|
| | SR ($\uparrow$) | PS ($\downarrow$) | SR ($\uparrow$) | PS ($\downarrow$) | SR ($\uparrow$) | PS ($\downarrow$) |
| SayCan | 27.55%$\pm$5.43% | 19.98$\pm$6.82 | 34.31%$\pm$4.80% | 21.43$\pm$3.31 | 45.11%$\pm$8.52% | 14.39$\pm$3.04 |
| LLM-Planner | 23.31%$\pm$4.58% | 20.74$\pm$5.83 | 39.07%$\pm$5.89% | 15.81$\pm$2.35 | 47.18%$\pm$8.42% | 16.33$\pm$2.79 |
| ExRAP | **52.75%$\pm$9.81%** | **13.36$\pm$3.62** | **54.89%$\pm$13.73%** | **10.59$\pm$3.84** | **55.12%$\pm$9.32%** | **10.07$\pm$2.61** |

**Limitation.** ExRAP leverages LLMs, which makes its performance dependent on the capabilities of these models to some extent. Compared to other baselines, it also requires increased computation effort due to the management of environmental context memory with temporal consistency. The ablation studies demonstrate that ExRAP is able to deliver robust performance even with a relatively lightweight LLM, yet further investigation into runtime overhead is desired.

# 6 Acknowledgement

This work was supported by Institute of Information & communications Technology Planning & Evaluation (IITP) grant funded by the Korea government (MSIT), (RS-2022-II220043 (2022-0-00043), Adaptive Personality for Intelligent Agents, RS-2022-II221045 (2022-0-01045), Self-directed multi-modal Intelligence for solving unknown, open domain problems, and RS-2019-II190421, Artificial Intelligence Graduate School Program (Sungkyunkwan University)), the National Research Foundation of Korea (NRF) grant funded by the Korea government (MSIT) (No. RS-2023-00213118), IITP-ITRC (Information Technology Research Center) grant funded by the Korea government (MIST) (IITP-2024-RS-2024-00437633, 10%), and by Samsung Electronics.

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

# A  Broader impact

Our work does not involve activities associated with negative societal impacts, such as disseminating disinformation, creating fake profiles, or conducting surveillance. Therefore, we do not expect any negative societal impacts from our research.

# B  Environment settings

## B.1  VirtualHome

We employ VirtualHome [8], a complex simulation environment designed for embodied AI research, which offers a wide range of interactive household activities. This environment requires an agent to perform tasks by interacting with various objects and following high-level action commands. VirtualHome features 162 different object types (e.g., TV, sofa) and multiple room types (e.g., kitchen, living room) across various indoor scenes, providing a complex environment through diverse combinations of rooms. Additionally, to standardize the time duration of skill execution, we have imposed the following restrictions on the 'walk' skill: walking is only possible to adjacent rooms, and only permitted towards objects that are present in that same room.

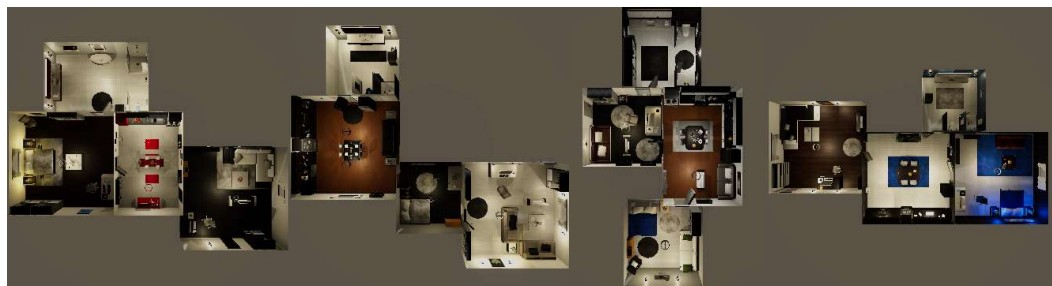

Figure A.1: Visualization of VirtualHome.

The embodied agent collects information about objects that come into its view and uses this as observations. Additionally, the agent utilizes seven skills to respond to the given continual instructions: walk *object* or walk *room*, grab *object*, switch *object*, put *object*, putin *object*, open *object*, and close *object*. For constructing TEKG, we use the graph based environment implemented in VirtualHome.

For non-stationarity, the environment condition involves a single continual instruction from the set changing at every predefined timestep: 4 for high non-stationarity, 6 for medium non-stationarity, and 8 for low non-stationarity. For continual instruction following tasks, we implement 19 continual instructions. Table A.1 shows the details of instructions.

## B.2  ALFRED

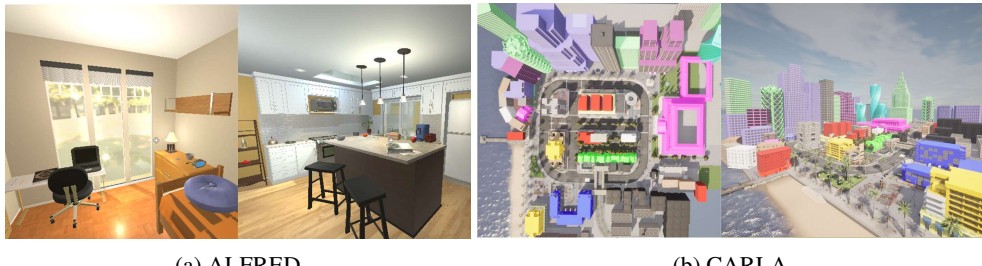

(a) ALFRED               (b) CARLA

Figure A.2: Visualization of ALFRED and CARLA.

We utilize ALFRED [9], which provides vision-and-language navigation and rearrangement tasks for embodied AI. This environment requires an agent to follow language formatted instructions to

Table A.1: Continual instructions in VirtualHome environment

| Example |
| --- |
| If no one is watching the TV, turn it on. |
| If you have an apple somewhere, bring it to your desk. |
| If you see a book somewhere unorganized, bring it to the sofa. |
| The dishwasher must always be open to dry the dishwasher. |
| It is good for maintenance if the microwave is always open. |
| Always leave the stove open. |
| The mug should always be on the coffeetable. |
| To wash dishes, place the plates in the microwave as shown. |
| If you see towels, put them in the washingmachine. |
| If your towel isn't stored somewhere else, put it in the closet. |
| If your computer stays off, turn it on. |
| If the cabinet is open, close it. |
| If someone reads a book and doesn't tidy it up, put it back. |
| If the stove is off, go and turn it on for preheat. |
| Put paper on the floor or anywhere else in the cabinet. |
| Place all visible mug in the microwave to sterilize them. |
| If the radio is off, turn it on |
| If someone uses a plate for washing dishes and leaves it somewhere, put it in the dishwasher. |
| If your microwave is off, turn it on. |

accomplish real-world-like household tasks. ALFRED features 58 different object types (e.g., bread) and 26 receptacle types (e.g., plate) across 120 various indoor scenes (e.g., kitchen).

The embodied agent gathers information about objects that enter its view and uses this as observations. Moreover, the agent utilizes six skills to respond to the given continual instructions: goto *object*, grab *object*, toggle *object*, put *object*, open *object*, and close *object*.

For non-stationarity, the environment condition involves a single continual instruction from the set changing at every predefined timestep: 4 for high non-stationarity, 6 for medium non-stationarity, and 8 for low non-stationarity. For continual instruction following tasks, we implement 16 continual instructions. Table A.2 shows the details of instructions.

Table A.2: Continual instructions in ALFRED environment

| Example |
| --- |
| If TV is off, turn it on. |
| If you have an apple somewhere, bring it to your coffeetable. |
| If you see a book somewhere unorganized, bring it to the sofa. |
| It is good for maintenance if the microwave is always open. |
| The mug should always be on the coffeetable. |
| To wash dishes, place the plates in the microwave as shown. |
| If your towel isn't stored somewhere else, put it in the closet. |
| If your computer stays on, turn it off. |
| If the cabinet is open, close it. |
| If someone reads a book and doesn't tidy it up, put it back. |
| Put paper on the floor or anywhere else in the cabinet. |
| Place all visible mug in the microwave to sterilize them. |
| If your microwave is on and spinning, turn it off. |
| If someone uses a plate for washing dishes and leaves it somewhere, put it in the sink. |
| If you see towels, put them in the sink. |
| If the alarmclock is on alone, turn it off |

## B.3   CARLA

CARLA is an opensource simulator for autonomous driving tasks that provides various environment settings for driving conditions and maps. For our experiment, we have modified CARLA to function

as a skill-based environment. The agent can navigate within the environment using three skills: turn right, turn left, and go straight. Additionally, the agent is capable of loading and offloading goods. The objective of the agent is to transport goods from a designated building to a target building whenever the conditions specified in the given instructions are met.

For non-stationarity, the environment condition involves a single continual instruction from the set changing at every predefined timestep: 4 for high non-stationarity, 6 for medium non-stationarity, and 8 for low non-stationarity. For continual instruction following tasks, we implement 16 continual instructions such as "If the green building calls, load the goods from the green building to the red building". We utilize a set of instructions as continual instructions.

## C    Implementation details

In this section, we provide the implementation details of our proposed framework ExRAP and each baseline. Our framework is implemented using Python v3.10 and trained on a system of an Intel(R) Core (TM) i9-10980XE processor and two NVIDIA RTX A6000 GPUs. Each experiment run takes 4 hours. We employ 4 different methods: ZSP [11], SayCan [1], ProgPrompt [3], and LLM-Planner [12]. Although the format differs, the observed environmental knowledge used remains consistent across all baselines. This includes observations of objects, object positions, object states, room adjacencies, etc. For generating a plan, we utilize the Llama-3 [37] model.

### C.1    Baseline

**ZSP** [11] capitalizes on the abilities of LLMs for embodied task planning by translating instructions into skill sequences, thereby enhancing the performance of embodied tasks. ZSP achieves this through the generation of detailed step-by-step prompts derived from examples of similar successful tasks, and then it utilizes the LLM to generate executable plans based on these examples. For implementation, we refer to the opensource [2].

**SayCan** [1] integrates the affordance function with language models, generating plans that are feasible to the context. SayCan achieves this by learning the environment affordance function derived from the LLM with the agent's experiences. Similarly, we calculate optimal affordance scores using environmental information and domain-specific knowledge. For observation, we utilize the linearized retrieved knowledge graph as the prompt for SayCan as same as ExRAP. For implementation, we refer to the opensource [3].

**ProgPrompt** [3] utilizes the code-style policy for generating plans for the embodied environment. As it passes the available primitive actions such as walk *object*, grab *object*, etc., in the form of `import` statements in Python, an available object list, and example tasks to LLM, the LLM produces some plans to succeed in the task. ProgPrompt can also execute conditional action by `assert-else` statements in the codes which is the output of the LLM. For experiments, we provide the code examples for various tasks in VirtualHome and ALFRED. For implementation, we refer to the opensource [4].

**LLM-Planner** [12] leverages the LLMs using the demonstrations with retrieved in-context examples, empowering embodied agents to perform complex tasks in environments with observed information, guided by natural language instructions. For our experiments, the LLM-Planner performs inference at each timestep, gathering observed environmental knowledge through natural language. When a skill execution fails, it captures the error and infers an alternative action based on the error at the next timestep. For observation, we utilize the detection results from the environment and incorporate them as prompts for the LLM-Planner. For implementation, we refer to the opensource [5].

The hyperparameter settings for baselines are summarized in Table A.3.

---

[2]https://github.com/huangwl18/language-planner
[3]https://github.com/google-research/google-research/tree/master/saycan
[4]https://github.com/NVlabs/progprompt-vh
[5]https://github.com/OSU-NLP-Group/LLM-Planner/

Table A.3: Hyperparameter settings for baselines

| Hyperparameters | Value |
|---|---|
| LLM | Llama-3-8B (Default) 
 Gemma-2B, Llama-3-70B (Ablation study) |
| Temperature | 0.33 |
| In-context example retriever | dpr-ctx_encoder-single-nq-base + BM25 |
| Number of prompts | 3 (LLM-Planner, ZSP) 
 2 (ProgPrompt) |
| Maximum new tokens | 40 |

## C.2 ExRAP

In ExRAP, we address multiple continual instruction-following tasks by decomposing each instruction into queries and task executions based on environmental context memory. To manage these tasks effectively, which are executed continuously and simultaneously, we introduce an exploration-integrated task planning scheme. This scheme incorporates information-based exploration into the LLM-based planning process, enhancing the balance between maintaining the validity of the environmental context memory and the demands of environment exploration, ultimately boosting overall task performance. Additionally, we implement a temporal consistency refinement scheme in our query evaluation to counteract the inherent decay of knowledge within the memory.

For the update function $\mu$ of TEKG in (3), we design an algorithmic refinement function tailored for embodied environments. This refinement function operates under two simple rules. First, only the most recent timestep data is retained for quadruple related agents (i.e., where the source entity or target entity is an agent). This is because information related to agents in the environment is observable. Second, semantically contradictory information is categorized into two cases: one where object states are opposite, such as simultaneous quadruples indicating that a TV is both off and on, and another where an object exists in two places at TEKG. Both scenarios result in the removal of the outdated quadruple.

The hyperparameter settings for the baselines are summarized in Table A.4.

Table A.4: Hyperparameter settings for ExRAP

| Hyperparameters | Value |
|---|---|
| LLM | Llama-3-8B (Default) 
 Gemma-2B, Llama-3-70B (Ablation study) |
| Temperature | 0.33 |
| In-context example retriever | dpr-ctx_encoder-single-nq-base + BM25 |
| Number of prompts | 3 |
| Number of retrieved quadruples $k$ for $\Phi_R$ | 12 |
| Number of iterations of query evaluator $\Phi_R$ | 10 |
| Maximum new tokens | 40 |
| Filtering threshold $\theta$ in (8) | 0.5 |
| Weights for exploration value $w_R$ in (12) | 1.0 |
| Weights for exploitation value $w_T$ in (12) | 0.01 |

# D Additional experiment

## D.1 Detailed results of non-stationarity and scale of continual instructions

Table A.5 presents a performance comparison in terms of SR and PS. The environmental settings consist of two variables: the size of continual instructions and the degree of non-stationarity. As the size of continual instructions increases, the number of instructions that need to be addressed also grows, requiring the agent to collect more knowledge and perform more tasks. Furthermore, a higher

---

**Algorithm 1** Detailed implementation of ExRAP framework

---
1: Continual instructions $\mathcal{I}$
2: Env. context memory $G_t$, Timestep $t$
3: Queries $\mathcal{Q}$, Executions $\mathcal{E}$
4: Instruction interpreter $\Phi_I$, Memory-augmented query evaluator $\Phi_M$
5: Exploitation planner $v_T$, Exploration planner $v_R$
6: $\mathcal{Q}, \mathcal{E}, C = \Phi_I(\mathcal{I})$
7: **loop**
8:     // (a) Query evaluation in (5)
9:     **for** $q$ in $\mathcal{Q}$ **do**
10:         $L_Q = [\,]$
11:         **for** 1, 2, ..., 10 **do**
12:             $\hat{G}_{1:t-1} \sim \Phi_R(G_{1:t-1}, \{q\})$
13:             $R(q|G_{t-1}) = \Phi_{\text{LLM}}\left(q, t, \hat{G}_{1:t-1}, P(q|G_{t-1})\right)$
14:             **if** hold (6) **then**
15:                 $L_Q$.append($P(q|G_{t-1})$)
16:             **end if**
17:         **end for**
18:         $R(q|G_{t-1}) = \mathbb{E}[L_Q]$
19:         $P(q|G_t) = \Phi_M(q, t, G_{1:t-1}, R(q|G_{t-1}))$
20:     **end for**
21:     // (b) Exploration-integrated task planning in (12)
22:     $z_t = \text{argmax}_{z \in Z}[w_T \cdot v_T(G_t, z) + w_R \cdot v_R(G_t, z)]$
23:     observation $o_{t+1} = \text{EnvStep}(z_t)$
24:     $G_{t+1} = \mu(G_t, o_{t+1})$, $t \leftarrow t + 1$
25: **end loop**

---

degree of non-stationarity means the environment changes more rapidly, necessitating that the agent focuses more on environmental information to adapt effectively.

As indicated in Table A.5, ExRAP demonstrates an increase in SR by $4.73\%$ to $27.50\%$ on average compared to the most competitive baseline, the LLM-Planner. Furthermore, ExRAP shows an average reduction in PS by $4.84$ to $13.29$ compared to the LLM-Planner. Similar to the experiments in the main text, ExRAP exhibits widening performance gaps as the complexity of tasks increases: the SR gap grows from $19.12\%$ compared to the LLM-Planner with small continual instructions, to $19.93\%$ with large continual instructions. Similarly, the PS gap expands from an average of $6.79$ to $9.97$. Moreover, ExRAP demonstrates robustness in embodied environments across varying levels of non-stationarity: the SR gap remains consistent, ranging from $17.77\%$ compared to the LLM-Planner in low non-stationarity environments to $17.32\%$ in high non-stationarity. The PS gap expands from an average of $6.96$ to $9.27$.

### D.2 Detailed results of ablation study

We compare the performance of ExRAP and ExRAP-TC, which evaluates queries without temporal consistency-based refinement, to understand the impact of this feature. Additionally, ExRAP-LLM operates by directly inputting the instruction "Explore the home" into the LLM, thereby enabling it to function as an exploration planner. This approach contrasts with ExRAP-EXP, which evaluates skills using an exploitation-only planner. Here, ExRAP-EXP inputs "Find {query}" for each query that is not yet satisfied, focusing solely on achieving specific task objectives without incorporating exploration. Table A.6 presents the detailed results of the ablation study for Tables 4 and 5 in Section 4.2 of the main text.

Table A.5: Performance comparison in VirtualHome

| Model | Low non-stationarity | | Medium non-stationarity | | High non-stationarity | |
|---|---|---|---|---|---|---|
| | SR ($\uparrow$) | PS ($\downarrow$) | SR ($\uparrow$) | PS ($\downarrow$) | SR ($\uparrow$) | PS ($\downarrow$) |
| **Small continual instruction. Num. of continual instructions is 3** | | | | | | |
| ZSP | 34.67%±10.01% | 15.68±1.40 | 33.11%±2.55% | 22.86±2.41 | 22.10%±3.59% | 24.84±12.50 |
| SayCan | 43.97%±10.71% | 16.58±2.75 | 40.58%±8.79% | 19.76±2.02 | 28.11%±6.80% | 21.13±10.20 |
| ProgPrompt | 43.40%±7.20% | 16.00±0.51 | 43.15%±3.22% | 19.99±1.59 | 30.71%±4.99% | 21.93±9.50 |
| LLM-Planner | 54.28%±6.47% | 19.54±4.19 | 42.61%±7.31% | 17.13±7.67 | 36.94%±9.93% | 20.91±6.97 |
| ExRAP | **71.78%±5.69%** | **12.35±1.63** | **67.77%±4.56%** | **12.29±0.96** | **51.67%±8.14%** | **12.58±2.72** |
| **Medium continual instruction. Num. of continual instructions is 5** | | | | | | |
| ZSP | 20.59%±4.71% | 31.03±4.68 | 20.06%±1.93% | 32.06±4.66 | 17.28%±3.16% | 24.08±4.63 |
| SayCan | 35.12%±4.83% | 21.67±3.81 | 33.69%±5.36% | 21.81±4.14 | 27.33%±4.24% | 16.18±3.98 |
| ProgPrompt | 32.10%±4.41% | 18.84±4.08 | 30.51%±5.31% | 23.43±1.07 | 27.19%±2.99% | 18.60±4.22 |
| LLM-Planner | 40.97%±7.00% | 17.61±1.40 | 39.89%±4.52% | 15.93±2.13 | 34.60%±6.49% | 14.94±2.89 |
| ExRAP | **61.12%±7.03%** | **11.75±2.49** | **55.14%±6.59%** | **11.33±1.92** | **50.12%±5.70%** | **8.61±2.25** |
| **Large continual instruction. Num. of continual instructions is 7** | | | | | | |
| ZSP | 14.69%±1.65% | 26.72±8.19 | 7.43%±6.2% | 59.16±26.04 | 10.32%±4.38% | 57.07±8.68 |
| SayCan | 32.50%±9.45% | 19.87±7.19 | 23.04%±12.26% | 37.05±17.43 | 24.20%±4.36% | 32.42±4.43 |
| ProgPrompt | 29.34%±3.16% | 20.89±5.60 | 21.20%±7.45% | 29.00±1.27 | 21.48%±6.48% | 40.05±6.19 |
| LLM-Planner | 41.10%±5.23% | 18.97±0.35 | 31.82%±14.30% | 17.63±2.34 | 26.39%±15.94% | 20.19±5.96 |
| ExRAP | **56.75%±8.19%** | **11.21±2.63** | **53.86%±8.59%** | **8.76±0.94** | **48.50%±7.43%** | **6.90±0.28** |

Table A.6: Detailed performance for ablation study

| Model | Low non-stationarity | | Medium non-stationarity | | High non-stationarity | |
|---|---|---|---|---|---|---|
| | SR ($\uparrow$) | PS ($\downarrow$) | SR ($\uparrow$) | PS ($\downarrow$) | SR ($\uparrow$) | PS ($\downarrow$) |
| **Small continual instruction. Num. of continual instructions is 3** | | | | | | |
| ExRAP-LLM | 41.38%±2.94% | 19.11±2.53 | 36.39%±11.49% | 21.45±1.68 | 33.60%±12.52% | 11.51±2.49 |
| ExRAP-MA | 46.22%±2.74% | 16.24±2.87 | 42.74%±9.52% | 18.41±2.97 | 36.00%±11.95% | 13.02±4.07 |
| ExRAP-TC | 49.82%±2.55% | 21.12±3.71 | 47.24%±7.55% | 23.37±2.26 | 40.41%±11.37% | **8.53±5.65** |
| ExRAP | **71.78%±1.69%** | **12.35±1.63** | **67.77%±4.56%** | **12.29±0.96** | **51.67%±8.14%** | 12.58±2.72 |
| **Medium continual instruction. Num. of continual instructions is 5** | | | | | | |
| ExRAP-LLM | 34.23%±24.57% | 15.17±5.11 | 29.42%±25.63% | 10.78±3.84 | 21.81%±14.06% | 14.65±2.40 |
| ExRAP-EXP | 43.33%±11.28% | 13.07±4.75 | 40.68%±9.77% | 13.03±3.01 | 29.36%±14.35% | 12.16±3.86 |
| ExRAP-TC | 47.25%±18.00% | 15.30±6.97 | 43.92%±7.97% | 15.47±2.17 | 27.91%±14.64% | 9.67±5.32 |
| ExRAP | **61.13%±13.76%** | **11.66±3.93** | **55.14%±6.59%** | **11.33±1.92** | **49.73%±8.88%** | **8.74±2.74** |
| **Large continual instruction. Num. of continual instructions is 7** | | | | | | |
| ExRAP-LLM | 40.88%±13.37% | 15.42±3.61 | 23.15%±12.71% | 12.27±3.32 | 21.51%±8.97% | 19.68±3.97 |
| ExRAP-MA | 43.87%±11.54% | 13.31±3.04 | 32.11%±10.70% | 10.98±2.45 | 23.46%±10.38% | 11.76±3.77 |
| ExRAP-TC | 46.21%±9.70% | **11.21±2.46** | 41.07%±8.69% | 9.69±1.57 | 26.40%±11.79% | 13.83±4.18 |
| ExRAP | **66.75%±8.19%** | **11.21±2.63** | **53.86%±8.59%** | **8.76±0.94** | **44.50%±7.43%** | **6.90±0.28** |

# E   Anaylsis

## E.1   Analysis of refinement temporal consistency

In Table A.7, we show examples of refined query responses. Although the input for subsequent query responses remains the same, the current timestep varies. Naturally, as timestep progresses, information decay should occur, and the entropy of the query response is expected to increase. However, the examples show an unexpected reduction in entropy from 0.24 to 0.15, leading to the removal of these outdated responses. This approach allows us to effectively model information decay, thereby improving the quality of query responses.

## E.2   Analysis of computation overhead of ExRAP

ExRAP employs sentence embedding techniques, such as DPR and BM25, for retrieving knowledge graphs and demonstrations to enhance exploitation value. As shown in the table below, the average retrieval and LLM inference time is approximately 14 times faster compared to inferring actions

Table A.7: Examples for temporal consistency-based refinement

| |
|---|
| Query: Tv is off |
| Timesteps: 42 |
| Environmental knowledge: (TV, inside, livingroom, 7), (TV, is, off, 7), (TV, is, on, 9), (TV, on, tvstand, 7) |
| (TV, is, off, 27), (kitchen, adjacent, bedroom, 1), (kitchen, adjacent, bathroom, 1) ... |
| Query response: Yes: 25%, No: 75% |
| Query: Tv is off |
| Timesteps: 43 |
| Environmental knowledge: (TV, inside, livingroom, 7), (TV, is, off, 7), (TV, is, on, 9), (TV, on, tvstand, 7) |
| (TV, is, off, 27), (TV, is, off, 33), (kitchen, adjacent, bathroom, 1) ... |
| Query response: Yes: 11%, No: 89% → Removed |

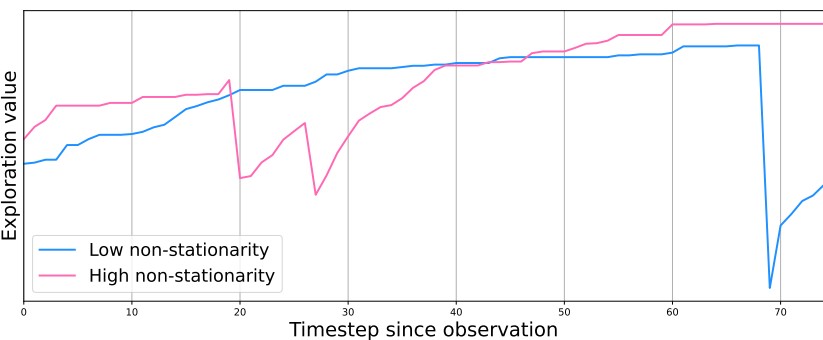

Figure A.3: Examples for exploration value w.r.t non-stationarity; we noted that the increase in exploration is more larger in environments with higher non-stationarity, leading to enhanced exploration. This, in turn, resulted in a more frequent drop in exploration value.

through an LLM without retrieval, using an RTX A6000 GPU and an i9-10980XE processor. By selecting relevant quadruples and demonstrations based on the query and instructions, ExRAP effectively reduces the context length, maintaining efficient LLM inference times. In our experiments, we retrieve only 3 demonstrations and 12 quadruples to generate prompts for each continual instruction, regardless of the size of the knowledge graph or dataset.

Table A.8: Comparison of Retrieval and LLM Inference Times

| | Retrieval (Quadruple) | Retrieval (Demos.) | LLM Inference | LLM Inference (w/o retrieval) |
|---|---|---|---|---|
| **Time (sec)** | 0.021 sec | 0.024 sec | 2.203 sec | 31.243 sec |

### E.3 Analysis of exploration dynamics

Figures A.3 illustrate the changes in exploration value for a single query during the execution of continual instructions. We observed that the exploration value increased steadily over time and decreased rapidly once information relevant to the query was collected. Specifically, we noted that the increase in exploration was more larger in environments with higher non-stationarity, leading to enhanced exploration. This, in turn, resulted in a more frequent drop in exploration value. As the reviewer suggested, investigating the exploration dynamics is an interesting analysis that demonstrates how ExRAP improves performance.

### E.4 Analysis of behavior of ExRAP

To analyze the behavior patterns of ExRAP, we map its generated plans to heuristic strategies such as Greedy, and Multiple Instruction First, and Max Staleness First. In our work, these heuristics are utilized only for analysis purposes, showing how ExRAP is able to achieve superior performance and how its exploration-integrated planning policy behaves differently in specific situations.

Table A.9: Examples for ExRAP behavior as Greedy heuristic

| Case |
| --- |
| Continual instructions: |
| 1. If you have an apple somewhere, bring it to your desk. |
| 2. If no one is watching the TV, turn it on. |
| 3. If your towel isn't stored somewhere else, put it in the closet. |
| Timesteps: 10 |
| Environmental knowledge: (TV, inside, livingroom, 7), (desk, inside, bedroom, 3), (apple, inside, kitchen, 9), |
| (apple, on, kitchencounter, 9), (kitchen, adjacent, bedroom, 1), (kitchen, adjacent, bathroom, 1), (towel, inside, bathroom) ... |
| Query response for instruction 1: Yes: 93%, No: 7% |
| Query response for instruction 2: Yes: 23%, No: 77% |
| Query response for instruction 3: Yes: 83%, No: 17% |

| Method | Plan (Step-wise inference) |
| --- | --- |
| ExRAP | walk kitchen, walk apple, grab apple, walk bedroom, walk desk, put apple desk, ... |
| Greedy | walk kitchen, walk apple, grab apple, walk bedroom, walk desk, put apple desk, ... |
| Multiple Instruction First | walk kitchen, walk apple, grab apple, walk bathroom, walk towel ... |
| Max Staleness First | walk bedroom, walk livingroom, walk tv, walk bedroom, walk kitchen, ... |

**Greedy.** When the entropy values of the queries are generally low and there are few executions required to complete, ExRAP operates in a manner similar to a greedy heuristic with respect to a single instruction. For example, in Table A.9, when there exist few executions required to complete and the entropy of the query responses is generally low, ExRAP operates similarly to a Greedy heuristic, executing instruction 1 independently. However, consistently applying this heuristic in all scenarios results in executing an instruction-wise plan, similar to the baselines.

Table A.10: Examples for ExRAP behavior as Multiple Instruction First heuristic

| Case |
| --- |
| Continual instructions: |
| 1. If you have an apple somewhere, bring it to your desk. |
| 2. If no one is watching the TV, turn it on. |
| 3. If your towel isn't stored somewhere else, put it in the closet. |
| Timesteps: 34 |
| Environmental knowledge: (TV, inside, livingroom, 2), (desk, inside, bedroom, 4), (apple, inside, kitchen, 5), (towel, inside, closet, 11) |
| (apple, on, kitchencounter, 3), (kitchen, adjacent, bedroom, 4), (kitchen, adjacent, bathroom, 7), (tv, is, off, 8) ... |
| Query response for instruction 1: Yes: 45%, No: 55% |
| Query response for instruction 2: Yes: 57%, No: 43% |
| Query response for instruction 3: Yes: 48%, No: 52% |

| Method | Plan (Step-wise inference) |
| --- | --- |
| ExRAP | walk kitchencounter, walk bedroom, walk desk, walk livingroom, walk tv ... |
| Greedy | walk bedroom, walk livingroom, walk tv, switch tv, ... |
| Multiple Instruction First | walk kitchencounter, walk bedroom, walk desk, walk livingroom, walk tv ... |
| Max Staleness First | walk bathroom, walk bathroomcounter, walk kitchen, walk kitchencounter, ... |

**Multiple Instruction First.** The Multiple Instructions First heuristic selects the skill based on the number of related instructions rather than focusing on a specific instruction. If the entropy values of various queries are generally high and there are many executions required to complete, ExRAP prioritizes skills where multiple instructions are concentrated. For example, in Table A.10, when the overall entropy of query responses is generally high, suggesting a need for efficient planning to explore multiple queries, ExRAP operates similarly to a Multiple Instructions First heuristic. However, consistently applying this heuristic in all cases can lead to an integrated plan, but result in failing to properly conclude tasks.

**Max Staleness First.** The Max Staleness First heuristic prioritizes selecting skills based on the age of information of related queries. Similarly, if the entropy value of a particular query is very high, ExRAP prioritizes exploring that specific query. This typically aligns with a Max Staleness First policy because our query evaluator operates with temporal consistency. For example, in Table A.11, when the entropy of query responses for instruction 2 is particularly high, indicating a need to explore such queries, ExRAP operates similarly to a Max Staleness First heuristic. While this heuristic focuses on specific queries and may reduce efficiency, it prevents the occurrence of instructions being left unexecuted, thereby avoiding starvation of certain instructions. This strategy essentially requires precise measurement of information decay to be effectively implemented. In ExRAP, we

Table A.11: Examples for ExRAP behavior as Max Staleness First heuristic

| Case |
| --- |
| Continual instructions: |
| 1. If you have an apple somewhere, bring it to your desk. |
| 2. If no one is watching the TV, turn it on. |
| 3. If your towel isn't stored somewhere else, put it in the closet. |
| Timesteps: 92 |
| Query response for instruction 1: Yes: 92%, No: 8% |
| Query response for instruction 2: Yes: 87%, No: 13% |
| Query response for instruction 3: Yes: 50%, No: 50% |
| Environmental knowledge: (TV, inside, livingroom, 2), (desk, inside, bedroom, 4), (apple, inside, kitchen, 89), (tv, is, off, 91) (apple, on, sink, 89), (kitchen, adjacent, bedroom, 4), (kitchen, adjacent, bathroom, 7), (towel, inside, bathroomcounter, 35) ... |

| Method | Plan (Step-wise inference) |
| --- | --- |
| ExRAP | walk kitchen, walk bathroom, walk bathroomcounter, walk faucet, ... |
| Greedy | walk livingroom, walk tv, switch tv, ... |
| Multiple Instruction First | walk livingroom, walk sofa, grab apple, switch tv, ... |
| Max Staleness First | walk kitchen, walk bathroom, walk bathroomcounter, walk faucet, ... |

perform temporal consistency-based refinement to enhance the quality of query responses, leading to improved performance.

As discussed so far, by appropriately utilizing different strategies including those similar to the three heuristics depending on given situations, our ExRAP dynamically adjusts its policy in the response to changes in the environment for multiple tasks of continual instruction following.

