# OpenReview forum: "Exploratory Retrieval-Augmented Planning For Continual Embodied Instruction Following"
_NeurIPS.cc/2024/Conference — NeurIPS 2024 poster_

### Official Review · Reviewer_kQXj · 2024-06-21

**Soundness:** 3
**Presentation:** 2
**Contribution:** 2
**Rating:** 6
**Confidence:** 4

**Summary:**

The paper introduces an Exploratory Retrieval-Augmented Planning (ExRAP) framework, that tackles the problem of embodied planning in dynamic environments. The study focuses on continual instruction following, which is when the task consists of multiple conditional subtasks. The execution of each subtask is dependent on the environment state. This requires continual exploration to keep up-to-date information and integrated planning abilities for efficiency.

ExRAP combines Large Language Models (LLMs) with environmental context memory represented as a temporal embodied knowledge graph (TEKG), that captures the states of objects in the scene and their pairwise relationships. The TEKG is updated at each step using the most recent observation and an update function $\mu$. Each subtask is translated into a pair of a query about the scene and a corresponding execution command with an instruction interpreter $\Phi_l$. Next, the memory-augmented query evaluator $\Phi_{M}}$ estimates the likelihood of the query being satisfied, using quadruples retrieved by $\Phi_R$ as environmental context information for the LLM. Authors enforce a temporal-consistency constraint on the evaluator, resulting in a continual steady decrease of the entropy of the answer. The execution commands corresponding to the satisfied queries are then passed to the execution part of the model. This part consists of exploitation and exploration planners. The former assesses the effectiveness of the skill in accomplishing the execution task. The latter one does the same with regard to reducing the uncertainty of the query evaluator $\Phi_M$. The performed skill is chosen as the one providing the maximum weighted sum of the two scores, balancing exploration and exploitation.

In the experiments, the authors measure the Success Rate (SR) of the task completion and the Pending Step (PS), representing the average number of steps taken to complete the task. The evaluation is performed on the VirtualHome, ALFRED, and CARLA environments. Performance in various degrees of stationarity in the environment is assessed, showing the advantage of the method. The performance with respect to the number of instructions is also evaluated. The method seems to excel in integrated planning, effectively solving multiple tasks with fewer steps. Ablation studies demonstrate the importance of temporal consistency and exploration-integrated planning. Intrestingly, the study reveals a limited effect of choosing a larger LLM as a base model.

**Strengths:**

- To the best of my knowledge, one of the first works to tackle conditional embodied planning in a dynamically changing environment.
- The results of the paper suggest that the method is clearly excels in producing more effective plans than the baselines, integrating multiple tasks in parallel. The plans are being adapted on the fly from the belief about the environment, and the belief is updated by neccessary exploration. This is a valuable contribution.

**Weaknesses:**

- The paper shows that the method is quite robust to the size of the base LLM. It could be a strength if there would be some training procedure possible. However, ExRAP relies exclusively on the in-context learning abilities of the LLM, thus the experiment shows that it scales poorly. The paper would benefit from an ablation study on the size of the retrieval dataset, so it can be shown that the method can scale with the context length of the model by increasing the number of examples.
- The method excels at following multiple simple parallel conditional instructions, but it is not clear if it will be as good at the long-term tasks that require multiple steps to achieve, like the ones on which the LLM-Planner was tested. The advantage of having an adaptable planner may become a disadvantage against the baselines, as it may require more queries to the environment context memory, which will create a large computational overhead. It is unclear whether the method is better than the baselines in integrating multiple complex tasks, each of which requires multiple steps, into an optimal plan. Unfortunately, this is not investigated in the paper.
- The applicability to the real-world setting is questionable and not investigated. Would the method scale appropriately with the number of objects in the scene? Methods like ConceptGraphs do not show robustness in that case.
- The termin "continual embodied instruction following" is a little misleading, as in the paper, the instructions are passed all at once, and not at random times, which would require replanning. The "conditional embodied instruction following in the dynamic environment" could be a better name, though this is not a significant issue.

**Questions:**

- Are the environment settings used in the in-context learning retrieval dataset the same or different from the ones used for the test?
- What does the observation $o_t$ consist of for ExRAP? Is it a list of quadruples? How would the setup be adapted for the real-world case?
- How does the system handles contradicting instructions? Like “If there is a storm outside, turn off the electricity in the house. If the TV says that the storm is over, turn the electricity on.”
- To me, it is not clear why the temporal consistency works. Would a simple baseline of reducing the temperature of prediction perform as good as the temporal consistency constraint in the paper?
- It would be interesting to investigate the exploration dynamics of the agent. Does the exploration scores on average decrease steadily throughout the experience or more rapidly in some particular times?

**Limitations:**

- Runtime overhead not investigated, but mentioned.
- Applicability to the real world is not addressed.

---

> ### Author Rebuttal · Authors · 2024-08-06
>
> Thank you for your thoughtful comments and insights.
>
> **W1** The scalablity of ExRAP.
>
> ExRAP uses the sentence embedding techniques such as DPR and BM25 for knowledge graph retriever and demonstration retriever for exploitation value.
> As shown in the table below, the average time taken for retrieval and LLM inference is about 14 times lower than the time it takes to infer actions through an LLM without retrieval, using an RTX A6000 GPU and an i9-10980XE.
>
> | Retrieval (Quadruple) | Retrieval (Demos.) | LLM Inference | LLM Inference (w\o retrieval) |
> |---|--|---|---|
> |0.021sec|0.024sec|2.203sec|31.243 sec|
>
> By selecting quadruples and demonstrations relevant to queries and instructions, it effectively reduces the context length and maintains the inference time of the LLM.
> In experiments, we retrieve only 3 demonstrations and 12 quadruples to generate prompts for each continual instruction, regardless of the size of the knowledge graph or the dataset.
> To clarify the retrieval models used, we will add these computational overhead results in Appendix.
>
> **W2** The complexity of the environment and computational overhead of ExRAP.
>
> For evaluation, we use continual instruction that involves multiple steps to achieve. For example, to address the single continual instruction "If you see a book somewhere unorganized, bring it to the sofa," the agent would need to execute the following steps:
>
> 0. initialized in livingroom
> 1. walk kitchen
> 2. walk bathroom
> 3. walk kitchen
> 4. walk bedroom (find book)
> 5. walk book
> 6. grab book
> 7. walk kitchen
> 8. walk livingroom
> 9. walk sofa
> 10. put book
>
> To effectively address complex continual instructions that require exploration, ExRAP performs continual instructions in an integrated approach. By doing so, ExRAP maximizes the effectiveness of the agent's actions for continual instructions, as evidenced by a reduction of 3.40 in the path length during experiments and resulting in a 16.45\% improvement in the success rate in experiments.
>
> In addition, as responded to in W1, ExRAP utilizes a retrieval-based approach to efficiently manage information. Even as the knowledge graph grows, it maintains the inference time of the LLM and minimizes computational overhead through retreival-based approach.
>
> **W3** The applicability to the real-world setting.
>
> The environments used in our experiments (VirtualHome, ALFRED, and CARLA) are highly similar to real-world settings and are frequently used in prior works proposing embodied planning or LLM-based frameworks [1, 2, 3]. For example, in VirtualHome, a single scene contains between approximately 20 to 100 objects, and there are 50 types of room structures, each with varying object states and positions.
>
> In ExRAP, a temporal embodied knowledge graph is used to efficiently manage and retrieve information on changing object states, enhancing adaptability and operational efficiency in dynamic environments.
>
> [1] Song, Chan Hee, et al. "Llm-planner: Few-shot grounded planning for embodied agents with large language models." Proceedings of the IEEE/CVF ICCV. 2023.
>
> [2] Singh, Ishika, et al. "Progprompt: Generating situated robot task plans using large language models." 2023 IEEE ICRA. IEEE, 2023.
>
> [3] Yang, Ruoxuan, et al. "Driving Style Alignment for LLM-powered Driver Agent." arXiv preprint (2024).
>
> **W4** Considering the term "continual embodied instruction following".
>
> Please see general response, Q2
>
> **Q1** Are the environment settings used in the in-context learning retrieval dataset the same or different from the ones used for the test?
>
> They are different. The demonstrations used in in-context learning and the test environments differ in the structure of rooms and positions of objects in VirtualHome and ALFRED, and the locations of buildings in CARLA.
>
> **Q2** What does the observation $o_t$ consist of for ExRAP? Is it a list of quadruples? How would the setup be adapted for the real-world case?
>
> Please see general response, Q1
>
> **Q3** How does the system handle contradicting instructions?
>
> In this paper, we focused on optimizing multiple continual instructions and did not address conflict resolution.  ExRAP can be expanded into a neural-symbolic approach [4] with a temporal embodied knowledge graph, utilizing various research to identify and resolve contradictions [5, 6].
> The question by the reviewer addresses a critical aspect of the multiple continual instructions and real-world implementation, and it is one of the future works for ExRAP. We include this content in the conclusion section to emphasize its importance.
>
> [4] Liu, Xiaotian et al. "A planning based neural-symbolic approach for embodied instruction following." Interactions 9.8 (2022): 17.
>
> [5] Li, Jierui et al.. "ContraDoc: Understanding Self-Contradictions in Documents with Large Language Models." arXiv preprint (2023).
>
> [6] Wan, Alexander et al. "What Evidence Do Language Models Find Convincing?." arXiv preprint (2024).
>
> **Q4** To me, it is not clear why the temporal consistency works. Would a simple baseline of reducing the temperature of prediction perform as well as the temporal consistency constraint in the paper?
>
> The problem of maintaining temporal consistency in LLMs is not resolved simply by lowering the temperature setting. As shown in Appendix D.1, LLMs often fail to reflect the increasing uncertainty of information over time, and this issue can occur even when the temperature is set to zero, particularly in small LLMs.
>
> To address this issue in LLMs, we employ repetitive reasoning and critic to enforce temporal consistency.
> This approach helps to ensure that the model maintains consistency over time, effectively capturing and reflecting temporal dynamics even in small LLMs.
>
> **Q5** It would be interesting to investigate the exploration dynamics of the agent. Do the exploration scores on average decrease steadily throughout the experience or more rapidly in some particular times?
>
> Please see the general response Q3, and PDF file

---

> > ### Author Response · Authors · 2024-08-13
> >
> > Thank you again for your considerate review. As the end of the author-reviewer discussion period approaches, we summarize the key points of our responses to the reviewer.
> >
> > 1. The scalablity of ExRAP
> > - ExRAP uses the sentence embedding techniques such as DPR and BM25 for knowledge graph retriever and demonstration retriever for exploitation value. Even as the knowledge graph grows, it maintains the inference time of the LLM and minimizes computational overhead through retreival-based approach. By selecting quadruples and demonstrations relevant to queries and instructions, it effectively reduces the context length and maintains the inference time of the LLM. In experiments, we retrieve only 3 demonstrations and 12 quadruples to generate prompts for each query and execution, regardless of the size of the knowledge graph or the dataset.
> > To clarify the retrieval models used, we will add these computational overhead results in Appendix.
> > 2. The applicability to the real-world setting.
> > - The environments used in our experiments (VirtualHome, ALFRED, and CARLA) are highly similar to real-world settings and are frequently used in prior works proposing embodied planning or LLM-based frameworks. For example, in VirtualHome, a single scene contains between approximately 20 to 100 objects, and there are 50 types of room structures, each with varying object states and positions.
> > - In ExRAP, a temporal embodied knowledge graph is used to efficiently manage and retrieve information on changing object states, enhancing adaptability and operational efficiency in dynamic environments.
> > 3. Investigatation for the exploration dynamics of the agent.
> > - Figures 1 in the PDF file illustrate the changes in exploration value for a single query during the execution of continual instructions. We observed that the exploration value increased steadily over time and decreased rapidly once information relevant to the query was collected. Specifically, we noted that the increase in exploration was more larger in environments with higher non-stationarity, leading to enhanced exploration. This, in turn, resulted in a more frequent drop in exploration value. As the reviewer suggested, investigating the exploration dynamics is an interesting analysis that demonstrates how ExRAP improves performance. We will add this to the Appendix of the final version.
> >
> > The reviewer's suggestion to analyze exploration dynamics offers valuable insight by demonstrating the characteristics of our agent's exploration, which helps readers' understanding. Additionally, the discussion on contradicting instructions points to potential approaches for resolving conflicts and offers constructive comments for our future work.
> >
> > We will do our best to address any questions the reviewer may have until the end of the reviewer-author discussion period. Please feel free to ask any further questions. Thank you.

---

> > > ### Comment · Reviewer_kQXj · 2024-08-13
> > >
> > > Thank you for your response. You addressed my concerns completely, and I will raise the score.

---

> > > > ### Author Response · Authors · 2024-08-13
> > > > **Thank you!**
> > > >
> > > > Thank you for raising your score. We greatly appreciate your thoughtful comments and feedback. We will include this discussion and further experiments in our final version.
> > > >
> > > > Your recommendation to analyze the exploration dynamics is insightful, as it clarifies how our agent investigates its environment, enhancing the reader's understanding. Furthermore, the discussion on contradicting instructions points to potential approaches for resolving conflicts and offers constructive comments for our future work.
> > > >
> > > > Thank you once again for your feedback. Our discussion is meaningful and highly motivating for us.

---

### Official Review · Reviewer_Jxmz · 2024-07-13

**Soundness:** 3
**Presentation:** 3
**Contribution:** 3
**Rating:** 7
**Confidence:** 3

**Summary:**

This paper presents the Exploratory Retrieval-Augmented Planning (ExRAP) framework to address the challenge of continual instruction following in non-stationary embodied environments. ExRAP enhances the reasoning capabilities of Large Language Models (LLMs) by efficiently exploring the environment and maintaining an environmental context memory to ground the task planning process.

The paper's main contributions are:

1.	A novel ExRAP framework integrating LLMs, memory-augmented reasoning, and information-based exploration for continual instruction following.
2.	Temporal consistency refinement and information-based exploration schemes tailored for ExRAP's integrated planning approach.

**Strengths:**

The memory-augmented query evaluation and exploration-integrated planning framework provide a principled way to bridge the gap between high-level language understanding and low-level embodied decision-making.

The temporal consistency refinement scheme is a novel contribution in embodied contexts that addresses the important issue of information decay in memory-based solutions over time. Using a temporal embodied knowledge graph (TEKG) as an environmental context memory enables the agent to efficiently assess the satisfaction of instruction conditions without constantly revisiting the environment.

**Weaknesses:**

1. ExRAP's performance still somewhat depends on the underlying LLM's reasoning abilities. While the ablation study shows robustness to smaller LLMs, it's unclear how the framework would perform with much weaker language models or in domains where the LLM's knowledge is limited.

2. The environments' non-stationary aspects are limited to a single instruction changing at fixed intervals. This does not fully capture the complexity and unpredictability of real-world non-stationary environments, where goals, conditions, and constraints can evolve in more intricate ways.

3. The instructions used in the experiments seem to be largely independent of each other, without any significant long-term dependencies or relationships between tasks. An example from Table A.1 is: "If no one is watching the TV, turn it on." If you have an apple somewhere, bring it to your desk. "If you see a book somewhere unorganized, bring it to the sofa." These instructions in the table appear to be independent tasks without any clear long-term dependencies or relationships between them. Each instruction can be completed independently of the others.

That being said, the ExRAP framework does take important steps by incorporating memory-augmented reasoning and exploration-integrated planning. These components provide a way for handling evolving instructions and non-stationary environments. The authors' claims may be somewhat overextended given the current experimental setup, but the work still makes valuable contributions to embodied instruction following.

**Questions:**

If authors can address points 2 and 3 from weaknesses, it will help in understanding whether there is a distinction between concurrent or multi-task instruction following and truly continual instruction following.

**Limitations:**

Authors have addressed some of the limitations of the work in the paper.

---

> ### Author Rebuttal · Authors · 2024-08-06
>
> Thank you for your thoughtful comments and insights.
>
> **W1** ExRAP's performance still somewhat depends on the underlying LLM's reasoning abilities. While the ablation study shows robustness to smaller LLMs, it's unclear how the framework would perform with much weaker language models or in domains where the LLM's knowledge is limited.
>
> When LLM performance degrades to a level where knowledge-based reasoning becomes unfeasible, a decline in ExRAP's performance is inevitable.
> However, as the reviewer commented, ExRAP demonstrates robust performance even in smaller LLMs, as shown in the ablation study (Section 4.2).
> If implementation with much weaker language models or in domains where the LLM's knowledge is limited is required, techniques such as imitation learning [1] or knowledge distillation [2] could be utilized to address these challenges.
>
> [1] Das, Abhishek, et al. "Embodied question answering." Proceedings of the IEEE conference on computer vision and pattern recognition. 2018.
>
> [2] Choi, Wonje, et al. "Embodied CoT Distillation From LLM To Off-the-shelf Agents." Forty-first International Conference on Machine Learning.
>
> **W2** The environments' non-stationary aspects are limited to a single instruction changing at fixed intervals. This does not fully capture the complexity and unpredictability of real-world non-stationary environments, where goals, conditions, and constraints can evolve in more intricate ways.
>
> At every change interval, the position or state of some objects related to an instruction is changed randomly. While the overall environment changes at fixed intervals, the specific objects that change at each interval are variable.
> This means that for each single continual instruction, the changes do not occur at fixed intervals. We will add these details comprehensively to the Appendix to clarify this aspect. Thank you for your comment.
>
> **W3** The instructions used in the experiments seem to be largely independent of each other, without any significant long-term dependencies or relationships between tasks. An example from Table A.1 is: "If no one is watching the TV, turn it on." If you have an apple somewhere, bring it to your desk. "If you see a book somewhere unorganized, bring it to the sofa." These instructions in the table appear to be independent tasks without any clear long-term dependencies or relationships between them. Each instruction can be completed independently of the others.
>
> We propose a framework that efficiently integrates and executes shared subtasks among continual instructions, even when each instruction appears to exist independently.
> For example, while moving to check if no one is watching TV, ExRAP can simultaneously verify the location of a book, or pick up a nearby apple and place it on the table while en route to turn on the stove.
>
> In response to the reviewer's comments, we conduct experiments on scenarios where query conditions explicitly overlap, or where the execution of one continual instruction explicitly impacts other instructions, such as "When computer is off, the mug should always be on the coffeetable, If your computer stays on, turn it off".
>
> | Model        | SR ($\uparrow$) | PS ($\downarrow$) |
> |--------------|-----------------|-------------------|
> | LLM-Planner  | 40.11%          | 15.80             |
> | ExRAP        | 51.36%          | 12.59             |

---

> > ### Comment · Reviewer_Jxmz · 2024-08-12
> >
> > Thank you for your detailed response. The response addressed my concerns.

---

> > > ### Author Response · Authors · 2024-08-12
> > > **Thank you!**
> > >
> > > Thank you once again for your insightful feedback and thoughtful comments.
> > >
> > > In particular, the discussion on continual instructions with long-term dependencies or relationships between them is invaluable.
> > > This results highlight our ExRAP's capability not only to handle independent continual instructions with implicit overlapping subtasks but also to perform effectively in scenarios where these instructions influence each other, emphasizing the generality and scalability of our approach.
> > >
> > > We will include these additional experimental results and the clarifications requested by the reviewers in the final version of our manuscript. Your review is constructive regarding our approach. Thank you!

---

### Official Review · Reviewer_K6e1 · 2024-07-14

**Soundness:** 3
**Presentation:** 3
**Contribution:** 3
**Rating:** 5
**Confidence:** 4

**Summary:**

The paper considers a problem of continual instruction following where an instruction contains multiple query-execution pairs to be performed.
For this, the paper proposes ExRAP comprised of two components: query evaluation and exploration-integrated task planning.
Query evaluation incrementally updates the temporal embodied knowledge graph and selects a set of the most relevant executions.
The exploration-integrated task planning part uses both exploration and exploitation by weight-summing the predicted values of predefined skills whose weights are hyperparameters.
The performance in the experiment shows noticeable margins compared to the baseline models.

**Strengths:**

- The paper is generally written well and easy to follow.
- The paper tackles an important and challenging problem of continual instruction following.
- Multiple benchmarks are explored for the proposed problem setup, implying generalizability of the setup, yet posing some scale issue (see weaknesses).

**Weaknesses:**

- While agreeing that addressing an instruction that contains multiple tasks, it is unclear why it should be in the explicit form of query(condition)-execution pairs. This is not well-justified. In addition, can't it be implicit?
- Regarding the first question, it looks like the proposed approach exploits the query-execution format of the proposed task setup (*e.g.*, query evaluator).
- The scale of the continual instruction following dataset is a bit limited. All the 3 datasets use less than 20 continual instructions, which may potentially raise a generalizability concern.
- High errors are observed across many tables in the experiments.
  - Why is this the case? Can this be related to the small scale of the dataset?
  - One argued contribution is a good performance of the proposed approach, but due to the high errors, it is unclear whether the argued outperformance is valid.
- The paper introduces many hyperparameters but it is unclear how much the proposed approach is sensitive to the choice of them.
- How to obtain the knowledge graph? is it obtained as GT values or predicted by some learned perception modules?
- In L184, why do the authors posit $H(R(q|G_{t-1})) > H(P(q|G_{t-1}))$? Any intuition behind this?

- Considering that the term "continual" also refers to learning some new tasks sequentially (*e.g.*, RL [a], IL [b], or even EIF [c]). Discussion about this aspect seems needed to clarify the usage of "continual" in this paper.

References\
[a] Xie and Finn. Lifelong robotic reinforcement learning by retaining experiences. CoLLAs, 2022.\
[b] Gao et al.  Cril: Continual robot imitation learning via generative and prediction model. IROS, 2021.\
[c] Kim et al. Online continual learning for interactive instruction following agents. ICLR, 2024

**Questions:**

See weaknesses above.

**Limitations:**

The authors do not address the potential negative societal impact.

---

> ### Author Rebuttal · Authors · 2024-08-06
>
> Thank you for your thoughtful comments and insights.
>
> **W1**. Should the continual instructions be explicit?
>
> The continual instructions are not always in the explicit form of query-execution pairs; they can also be implicit, such as "Always leave the stove open" in Appendix A.1. In that case, they can be interpreted and executed conditionally, "Query: stove is closed, Execution: open the stove"  by an instruction interpreter. Additionally, In our main text, within section 4.1 on Instruction type (Lines 276-284), we demonstrated that tasks could still be performed with minimal performance degradation even when multiple continual instructions are summarized or when objects are ambiguous.
>
> **W2** Exploitation of the query-execution format.
>
> We utilize an instruction interpreter to transform continual instructions, in which each instruction is either implicit or explicit, into a query-execution format. To address various format of continual instructions, we use LLM as the instruction interpreter.
> In the fields of real-time databases, event-based query processing, and monitoring system applications, it is common to separate user requests into query and execution components [1, 2].
> Inspired by these processing methods, we proposed an ExRAP framework that uses the query-execution structure for continual instructions.
>
> [1] Shaikh, Salman Ahmed, et al. "Smart query execution for event-driven stream processing." 2016 IEEE Second International Conference on Multimedia Big Data (BigMM). IEEE, 2016.
>
> [2] Liu, Ling, Calton Pu, and Wei Tang. "Continual queries for internet-scale event-driven information delivery." IEEE Transactions on Knowledge and Data Engineering 11.4 (1999): 610-628.
>
> **W3** The scale of the continual instructions.
>
> For evaluation, we construct the continual instructions by sampling 3, 5, or 7 instructions from a set of 19 continual instructions.
> Choosing 5 of these 19 possible continual instructions results in 11,628 unique combinations.
> We also evaluate the types of instructions to demonstrate the generalization performance across different scenarios, ensuring that our approach can effectively handle a diverse range of instructions.
>
> **W4** High errors are observed across many tables in the experiments.
>
> The time-varying features of the environment and the large number of combinations of continual instructions contribute to a large variance in performance.
> Despite this, the experimental results remain valid. As shown in Table 1 of the main text, the confidence intervals for the experiments in the ALFRED and CARLA environments do not overlap with those of the baselines, indicating that our results are statistically significant.
> To minimize experimental error, we will add to increase the random seeds from 5 to 10 used in the experiments.
> The below table shows the performance of the ExRAP and other baselines with 10 random seeds in medium non-stationarity, VirtualHome.
> As the error bars decrease sufficiently and the overlap with the baselines disappears, we can observe that the results become statistically significant.
>
> | Model         | SR ($\uparrow$)           | PS ($\downarrow$)      |
> |----|-----|------|
> | ZSP           | 20.06% ± 1.93%            | 32.06 ± 4.66           |
> | SayCan        | 33.69% ± 5.36%            | 21.81 ± 4.14           |
> | ProgPrompt    | 30.51% ± 5.31%            | 23.43 ± 1.07           |
> | LLM-Planner   | 39.89% ± 4.52%            | 15.93 ± 2.13           |
> | ExRAP     | 55.14% ± 6.59%            | 11.33 ± 1.92           |
>
> **W5** Hyperparameters sensitivity.
>
> In our experiments, the hyperparameters that impact performance are the exploration value coefficient $w_R$ and the exploitation value coefficient $w_T$.
> The table below exhibits robust performance across a wide range of hyperparameter settings.
> When the exploitation value coefficient is high, exploration may be reduced, leading to a slight decrease in performance.
> However, the advantage of integrated planning across multiple instructions still results in comparable performance.
> When the exploration value coefficient is high, an excessive focus on gathering information can also lead to a slight decrease in performance.
> We will add these results to the Appendix for the final version.
>
> | $w_R: w_T$ | SR ($\uparrow$) | PS ($\downarrow$) |
> |------------|-----------------|-------------------|
> | 1000:1     | 49.70%          | 14.67             |
> | 100:1      | 55.14%          | 11.33             |
> | 10:1       | 51.33%          | 13.98             |
> | 1:1        | 42.62%          | 16.66             |
>
> **W6** How to obtain the knowledge graph? is it obtained as GT values or predicted by some learned perception modules?
>
> Please see general response, Q1
>
> **W7** The intuition of $H(R(q|G_{t-1})) > H(P(q|G_{t-1}))$.
>
> This is based on the intuition that the uncertainty of information is increased over time in time-varying environments.
> For instance, if the location of an object is observed at step $t-1$, the uncertainty about the object's location may stay the same (not moved) or increase by step $t$ in a non-stationary environment.
> Consequently, the uncertainty associated with the knowledge graph $G_{t-1}$ increases monotonically by the time it reaches step $t$. Therefore, when evaluating queries based on this knowledge graph, the uncertainty in the query response also increases. This approach is the same as common practices in fields such as sensor databases and monitoring systems, where dealing with data uncertainty is crucial [3, 4].
>
> The formula should be corrected to $ H(R(q|G_{t-1})) \geq H(P(q|G_{t-1}))$. Thank you for detailed review.
>
> [3] Prasad Sistla, A., et al. "Querying the uncertain position of moving objects." Temporal databases: research and practice (1998): 310-337.
>
> [4] Cheng, Reynold, and Sunil Prabhakar. "Managing uncertainty in sensor database." ACM SIGMOD Record 32.4 (2003): 41-46.
>
> **W8** Considering the meaning of the word "continual."
>
> Please see general response, Q2

---

> > ### Comment · Reviewer_K6e1 · 2024-08-11
> > **Response by Reviewer K6e1**
> >
> > Thank you for your detailed response. The response addressed my concerns but a few things still remain unclear.
> > - For W1, "Always leave the stove open" seems not implicit but explicit with the condition that is always `True`. The current continual instruction proposed is usually in the form of condition-execution pairs (i.e., if something is happening, then do something). My concern is why we need the "two", condition and execution. For instance, can't it be done with only execution?
> > - For W8, thank you for the clarification, but this can be quite confusing and it might be better to clarify this somewhere in the main paper.

---

> ### Author Response · Authors · 2024-08-11
>
> Thank you for your kind feedback and the additional questions. Our responses are as follows:
>
> **For W1.**
>
> - Thank you for valuable insight. Our ExRAP can also be applied in situations where instructions consist solely of execution. Below table shows the performance of simple experiments on instructions that involve only execution and no conditions. Although the need for exploration has been reduced, ExRAP demonstrates an advantage over the baseline by efficiently executing multiple instructions through integrated task planning. We will include this results and experiments comparing the entire baselines in the Appendix of the final version.
>
> | Model | SR (↑) | PS (↓) |
> | --- | --- | --- |
> | LLM-Planner | 74.17% | 8.80 |
> | ExRAP | 79.86% | 7.20 |
>
> - As the reviewer mentioned, in environments where only the robot agent exists and there are no changes over time, the condition of "Always leave the stove open" can be True. However, in dynamic environments where other agents or users can interact with the stove, the instruction would need to be divided into a condition and execution: "If the stove is closed", then "open the stove". For instructions that appear to only involve execution, such as "put the mug in the sink",ExRAP addresses this by relating the condition to the goal, such as “if the mug is not in the sink". If we treat the condition as always true, it could force the repetition of an already completed task, potentially leading to performance degradation. These examples are considered important for explaining our framework, and we will ensure to include them in our paper.
> - Converting to a query-execution format becomes a crucial part in optimizing exploration in ExRAP.
> However, simply separating query and execution does not fully resolve the continual instruction following problem in time-varing dynamic environment. We manage environmental observations through an knowledge graph-based approach, and efficiently utilize information through retrieval to evaluate queries. Additionally, an integrated exploration and exploitation planner enables efficient planning through multiple instructions.
>
> **For W8.**
> - Thank you for your thoughtful feedback. We will add the following statement to the main text, Line 108:
>
> “The continual instruction does not refer to continual learning, which learns some new task sequentially.
> It aligns closely with the concept of continual query, which are standing queries that monitor update of interest and return results whenever the update reaches specific thresholds.”
>
> We are very pleased that the reviewer's concerns and most of the identified weaknesses have been addressed. If you have any additional questions, please feel free to ask. Thank you!

---

> > ### Comment · Reviewer_K6e1 · 2024-08-12
> > **Thank you for the response.**
> >
> > I thank the authors for the response with the additional experiment. The response addressed my concerns and I am happy to increase the score.

---

> > > ### Author Response · Authors · 2024-08-12
> > > **Thank you!**
> > >
> > > Thank you for raising your score. We truly appreciate your consideration and constructive comment and feedback. We will incorporate this discussion and additional experiments into the final version.
> > >
> > > While our paper focus on solving continual instruction, your suggestion to extend our experiments to include what appears to be purely execution-based instruction is insightful feedback that can demonstrate the generality of our model.
> > >
> > > Thank you once again for your response. The discussion with you is very meaningful and greatly encourages us.

---

### Author Rebuttal · Authors · 2024-08-06

**General Response**

We thank the reviewers for their valuable feedback. We are encouraged that they found our approach is novel, well-written, easy to follow, adresses the important and challenging problem, and provides valuable contribution,  as well as providing extensive evaluations which demonstrate the effectiveness of our approach in multiple benchmarks. We addressed all weaknesses and questions to resolve the concerns raised. For those areas where the reviewer requested additional information, we conducted further studies and provided detailed explanations to enhance overall understanding.

**Q1**  Consideration the observation of environment. (Reviewer K6e1, Reviewer kQXj)

In alignment with prior works on LLM-based agents [1, 2, 3, 4], the agent collects text-based observations about objects visible within its field of view or through interaction with objects.
It is then simply transformed into quadruples (e.g., "book is on the table" becomes ("book", "on", "table")). ExRAP constructs a temporal embodied knowledge graph using these observations.

However, addressing the reviewer's concern, we conducted experiments under the assumption that the agent collects observations via camera and utilizes an object detection module or Vision-Language Models (VLMs) for text observation, resulting in only partial information being detected when observing the environment.
Below table demonstrates that even with a decrease in detection probability, sufficient performance is maintained.

| Detection Prob. | SR ($\uparrow$) | PS ($\downarrow$) |
|-----------------|-----------------|-------------------|
| 80%             | 50.77%          | 13.52             |
| 90%             | 52.97%          | 12.38             |
| 95%             | 54.71%          | 11.84             |
| 100%            | 55.14%          | 11.33             |

[1] Song, Chan Hee, et al. "Llm-planner: Few-shot grounded planning for embodied agents with large language models." Proceedings of the IEEE/CVF International Conference on Computer Vision. 2023.

[2] Xiang, Jiannan, et al. "Language models meet world models: Embodied experiences enhance language models." Advances in neural information processing systems 36 (2024).

[3] Lin, Bill Yuchen, et al. "On grounded planning for embodied tasks with language models." Proceedings of the AAAI Conference on Artificial Intelligence. Vol. 37. No. 11. 2023.

[4] Singh, Ishika, et al. "Progprompt: Generating situated robot task plans using large language models." 2023 IEEE International Conference on Robotics and Automation (ICRA). IEEE, 2023.

**Q2**  Consideration the term "continual embodied instruction following". (Reviewer K6e1, Reviewer kQXj)

As noted in the main text (Lines 100-108), "continual instruction" is defined as multiple instructions that need to be performed continuously and simultaneously. So, the term "continual" as we use it does not refer to continual learning, which learns some new task sequentially. Rather, this usage aligns closely with the concept of continual query [6, 7], which are standing queries that monitor update of interest and return results whenever the update reaches specific thresholds.
Therefore, continual instruction following refers not merely to following conditional instructions but to a broader problem definition. It includes the need to continuously execute instructions in response to the environment and to simultaneously address multiple instructions.

**Q3**  It would be interesting to investigate the exploration dynamics of the agent. Do the exploration scores on average decrease steadily throughout the experience or more rapidly in some particular times? (Reviwer kQXj)

Figures 1 in the PDF file illustrate the changes in exploration value for a single query during the execution of continual instructions.
We observed that the exploration value increased steadily over time and decreased rapidly once information relevant to the query was collected. Specifically, we noted that the increase in exploration was more larger in environments with higher non-stationarity, leading to enhanced exploration. This, in turn, resulted in a more frequent drop in exploration value. As the reviewer suggested, investigating the exploration dynamics is an interesting analysis that demonstrates how ExRAP improves performance.
We will add this to the Appendix of the final version. Thank you for the suggestion.

**Q4** The potential negative societal impact.

Our work does not involve activities associated with negative societal impacts, such as disseminating disinformation, creating fake profiles, or conducting surveillance. Therefore, we do not expect any negative societal impacts from our research.

We will add this statement explicitly in the final version.

---

### Decision · Program_Chairs · 2024-09-25

**Decision:**

Accept (poster)

**Comment:**

This paper presents the Exploratory Retrieval-Augmented Planning (ExRAP) framework to address the challenge of continual instruction following in non-stationary embodied environments. ExRAP enhances the reasoning capabilities of Large Language Models (LLMs) by efficiently exploring the environment and maintaining an environmental context memory to ground the task planning process. The paper provides insights into integrating LLMs, memory-augmented reasoning, and information-based exploration for continual instruction following. Authors have also replied and satisfactorily replied to the concerns raised by the reviewers during the rebuttal period. Overall a good paper.